# Origamic metal-organic framework toward mechanical metamaterial

Eunji Jin[1], In Seong Lee[1], D. ChangMo Yang ⬤[1], Dohyun Moon ⬤[2], Joohan Nam[1], Hyeonsoo Cho[1], Eunyoung Kang[1], Junghye Lee[1], Hyuk-Jun Noh[1], Seung Kyu Min ⬤[1,3] ✉ & Wonyoung Choe ⬤[1,4] ✉

Origami, known as paper folding has become a fascinating research topic recently. Origami-inspired materials often establish mechanical properties that are difficult to achieve in conventional materials. However, the materials based on origami tessellation at the molecular level have been significantly underexplored. Herein, we report a two-dimensional (2D) porphyrinic metal-organic framework (MOF), self-assembled from Zn nodes and flexible porphyrin linkers, displaying folding motions based on origami tessellation. A combined experimental and theoretical investigation demonstrated the origami mechanism of the 2D porphyrinic MOF, whereby the flexible linker acts as a pivoting point. The discovery of the 2D tessellation hidden in the 2D MOF unveils origami mechanics at the molecular level.

Paper folding, known as origami, is no longer limited to craft activities[1–3]. Origami design principles are now extended to art[4], science[5], engineering[6], architecture[7], and further to industry[8,9], because of the fascinating deployable nature of origami architectures, despite the origins of materials used for their construction. The list of origami applications in technology is rapidly growing, as exemplified by solar cells[10], foldable and flexible electronics[11], lithium-ion batteries[12], and biomedical devices[13–15]. The length scales used for origami have also evolved, ranging from the meter to the nanoscale[16–20]. These recent origami activities are close to related known origami tessellations, such as Miura-ori[21,22], double corrugation surface (DCS)[23,24], Ron Resch[25], waterbomb[26], Yoshimura[27], and square twist patterns[28] (Supplementary Fig. 1). Each of the origami tessellations consists of the same or different repeating patterns. Interestingly, a folding mechanism can be changed by the valley-mountain fold despite having the same repeating pattern. For example, the DCS and square twist patterns exhibit the same repeating patterns, but their folding movement differs. Both tessellations are highly deployable[29,30] and can serve as a blueprint for constructing mechanical metamaterials with negative Poisson's ratio, which is well-known for an exotic mechanical property[31,32]. Despite the advent of various origami-inspired materials, a daunting challenge has been to build molecular materials based on origami tessellations.

To create origami-inspired materials at the molecular level, MOFs could serve as an ideal platform for mimicking origami patterns, thanks to the unique features that the building blocks, metal nodes, and organic linkers, used for MOF construction are virtually limitless and exquisitely tunable[33,34]. Through rational design based on deformable net topology, many MOFs have exhibited structural flexibility, derived from the inherent flexibility of their structural building blocks over the past two decades[35–37]. The rich structural choices serve to realize the deployable 2D framework itself, showing a property like a negative thermal expansion[38]. While the predictable deployable movement of these flexible MOFs demonstrates mechanical properties with the metamaterials[39–42], a geometrical analysis involving origami tessellation to uncover hidden dynamic motions in MOFs beyond typical topological analysis is still in its infancy[43].

Here, we report a MOF based on DCS origami tessellation, assembled from a flexible porphyrin linker, and a Zn paddlewheel secondary building unit (SBU). The thermal movement unveiled in this MOF is controlled by DCS origami mechanics, exhibiting unusual folding behavior—an origami movement demonstrated in framework

[1]Department of Chemistry, Ulsan National Institute of Science and Technology, 50 UNIST, Ulsan 44919, Republic of Korea. [2]Beamline Department, Pohang Accelerator Laboratory, Pohang, Republic of Korea. [3]Center for Multidimensional Carbon Materials (CMCM), Institute for Basic Science (IBS), Ulsan 44919, Republic of Korea. [4]Graduate School of Carbon Neutrality, Ulsan National Institute of Science and Technology, Ulsan 44919, Republic of Korea. ✉e-mail: skmin@unist.ac.kr; choe@unist.ac.kr

solids. We expect that such MOFs based on origami tessellation can be actively utilized as an emerging class of mechanical metamaterials in the near future.

## Results

### PPF-301

PPF-301 crystals were synthesized with $Zn(NO_3)_2 \cdot 6H_2O$ and 5,10,15,20-tetrakis [4-carboxymethyleneoxyphenyl] porphyrin (TCMOPP) (Supplementary Fig. 2) via a solvothermal reaction. The as-synthesized PPF-301 crystals display a pale purple color and exhibit a rectangular plate shape (Supplementary Fig. 3). PPF-301 consists of Zn paddlewheel SBUs and ZnTCMOPP in a 2 to 1 stoichiometric ratio (Fig. 1A), forming 2D layers. During the reaction, the porphyrin core in TCMOPP undergoes metallation, resulting in a five-coordinate Zn ion that coordinated the DMF solvent. The functionalized aryloxy group of the porphyrin backbone allows for two different orientations, forming the Zn SBUs. Two DMF solvents coordinate with the exterior axial position of each Zn SBUs (Supplementary Fig. 4). Notably, the self-assembled 2D layer of PPF-301 exhibits a corrugated structure due to the flexible aryloxy groups in the TCMOPP linker, which is in contrast to another porphyrinic MOF, PPF-1, where 2D square grids are built from a rigid tetratopic porphyrinic linker[44]. The synchrotron powder X-ray diffraction pattern of the as-synthesized PPF-301 matches well with the simulated pattern (Supplementary Fig. 5) and shows an isostructure with a 2D porphyrinic MOF reported by the Goldberg group[45]. The 2D layers in PPF-301 are alternately stacked in a stepwise fashion with an interlayer distance of 7.8 Å, parallel to $(\bar{1}11)$ plane of the crystal (Supplementary Fig. 6). In this stacking system, free DMF solvents are observed between the 2D layers. The presence of solvents in the framework

contributes to the interaction between interlayers, maintaining a close interval. The coordinated DMF molecules were observed using FT-IR spectra in addition to crystallographic data (Supplementary Fig. 7). The solvent content of the as-synthesized PPF-301 was confirmed through a $^1H$ NMR experiment. The ratio of TCMOPP to DMF was found to be 1.08:5.04, which is similar to the ratio obtained from the crystallographic data (Supplementary Fig. 8). PPF-301 is thermally stable up to ~700 K and non-porous to $N_2$ at 77 K (Supplementary Figs. 9 and 10). Small amounts of $CO_2$ were adsorbed at 195, 273, and 298 K to 4.26, 1.96, and 1.77 mmol/g, respectively (Supplementary Fig. 11).

When we attempted to simplify the 2D pattern of PPF-301, by connecting nodal points as the oxygen of the aryloxy group $(-OCH_2-)$ in TCMOPP as shown in Fig. 1B, we were surprised to find that the 2D pattern of PPF-301 is closely related to that of the DCS origami tessellation. The analytic methodology based on flexible points is different from topological analysis, which is defined in the MOF field. The 2D layer in PPF-301 can be analyzed as an hourglass tessellation[46], connecting the centroid of the Zn cluster with the Zn atom in the core of porphyrinic ligands including the oxygen of the aryloxy group. However, as changing viewpoints for structural simplification, we found fascinating patterns. DCS origami tessellation, found in the 2D layer, consists of repeated mountain and valley patterns as trigonal shapes. Because the DCS pattern is highly deployable, we hypothesized that the PPF-301 could exhibit origami movement if these nodal points are flexible.

### Thermal response and origami tessellation of PPF-301

To test a possible structural change in PPF-301, we performed temperature-dependent synchrotron single-crystal X-ray diffraction (SCXRD) at the Pohang Accelerator Laboratory, South Korea

**A**

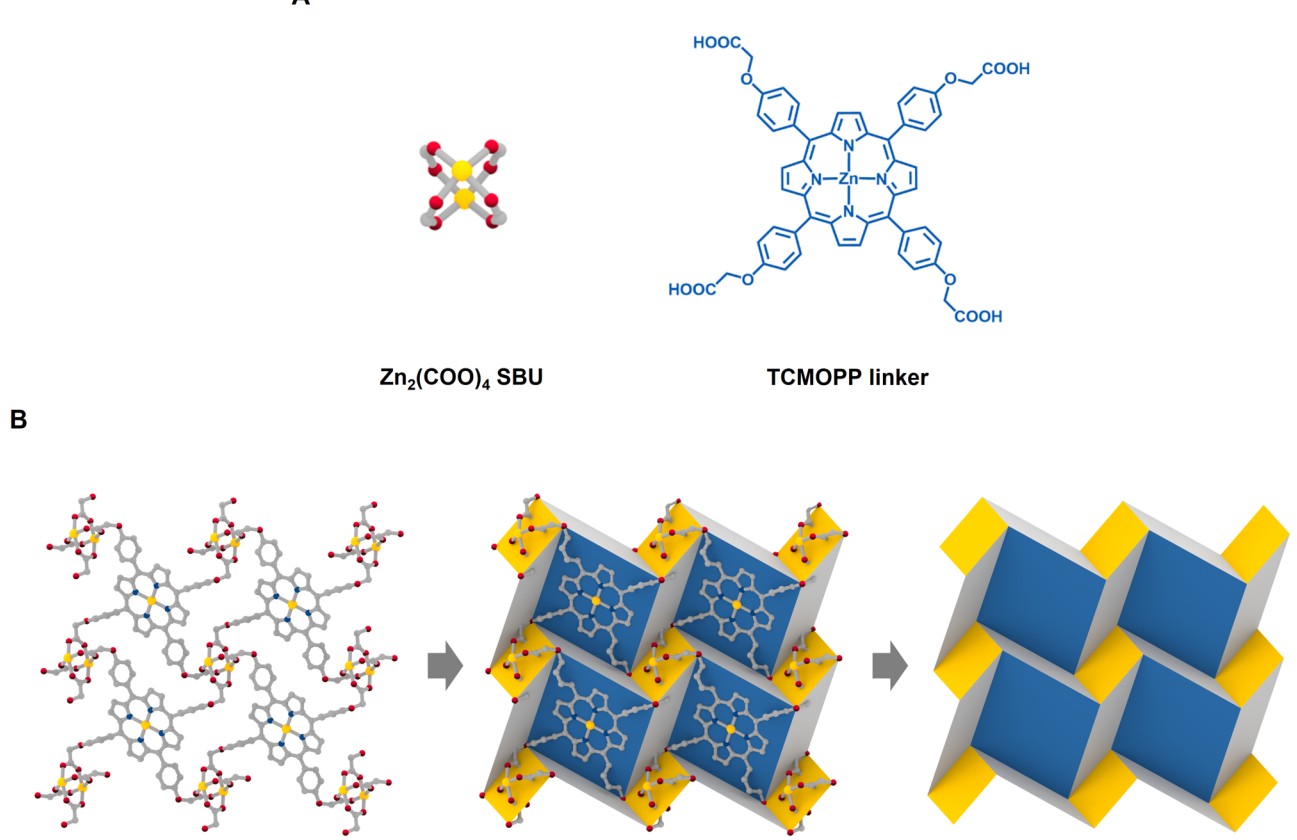

Zn₂(COO)₄ SBU                          TCMOPP linker

**B**

**Fig. 1 | Crystal structure and origami tessellation unveiled in PPF-301. A** Two building blocks for PPF-301: $Zn_2(COO)_4$ SBU and TCMOPP linker. Zn = yellow; C = gray; N = blue; O = red; all hydrogen atoms and solvent molecules are omitted for clarity. **B** Simplification of 2D porphyrinic MOF, leading to origami tessellation. Solvents and hydrogen are omitted for clarity. The blue and yellow tiles filled the TCMOPP linker and Zn SBU, respectively. Red balls are oxygen atoms of the aryloxy group.

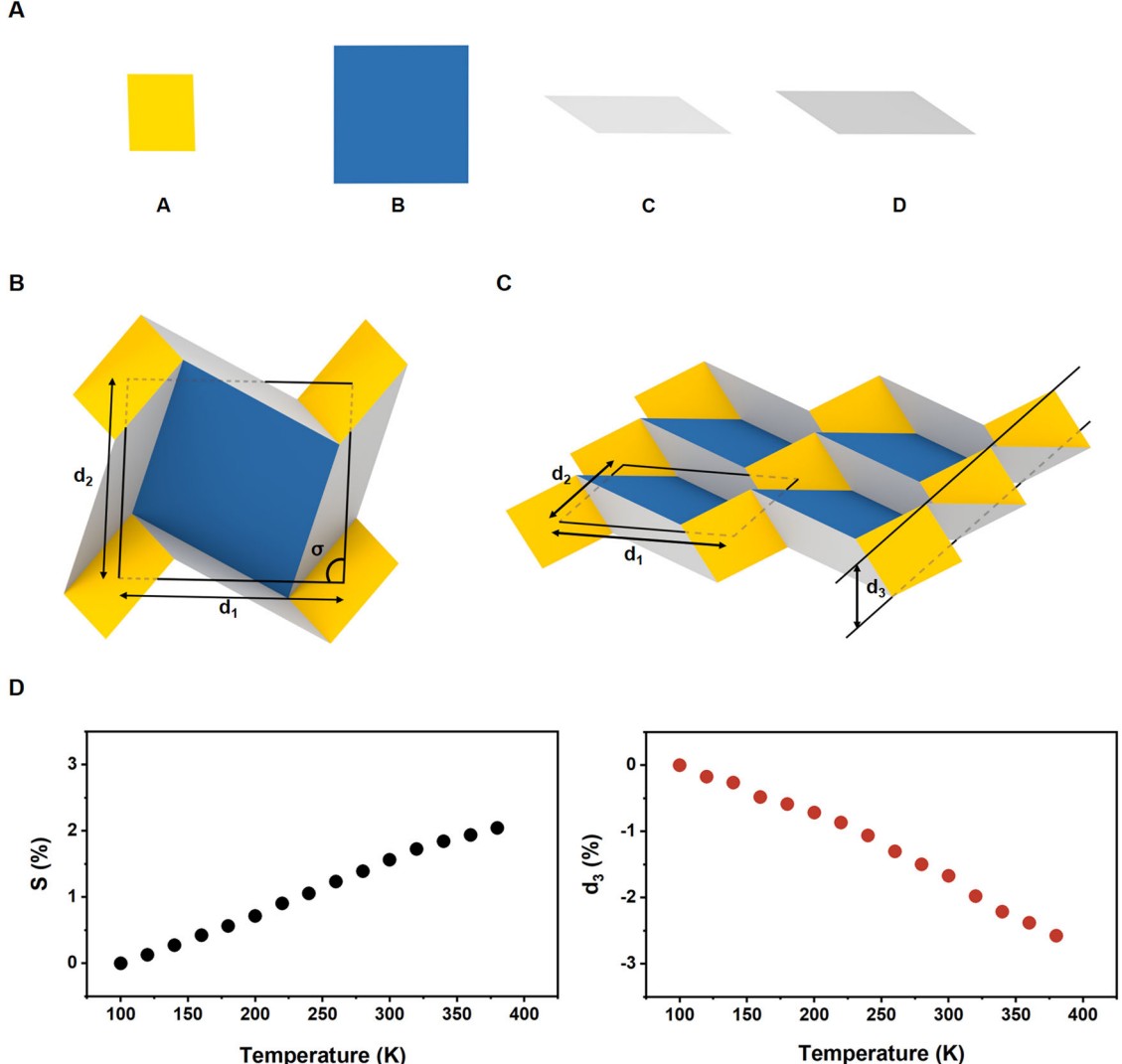

**Fig. 2 | Thermal response of PPF-301. A** Four types of different-sized tiles, filled in the 2D sheet. A; Zn SBU (yellow), B; porphyrinic ligand (blue), C and D; hollow tiles (gray and dark gray, respectively). **B** 2D area ($S$) of the PPF-301. Area $S$ is defined by connecting each centroid of Zn SBUs ($S = d_1 \times d_2 \times \sin\sigma$). **C** Thickness ($d_3$) of the 2D corrugated layer. **D** $S$ and $d_3$ as a function of temperature from 100 to 380 K. Source data are provided as a Source Data file.

(Supplementary Table 1). During the experiment conducted over an extended period of time, we prepared a crystal in a sealed capillary, including a small amount of solvent to prevent any loss of crystallinity. Firstly, we note an interesting change in the cell parameters of PPF-301 in a temperature range of 100–380 K. As the temperature decreases from 380 to 220 K, the cell parameters exhibit complete reversibility without hysteresis (Supplementary Fig. 12). The cell volume progressively increases by 5.2% upon heating, accompanied by changes in the $a$ and $b$ parameters, as well as the γ value (Supplementary Fig. 13). To analyze such structural changes in detail, we focus on the 2D area ($S$) and interlayer spacing of PPF-301. The area $S$ and the interlayer spacing increase by 2.0% and 3.1%, respectively (Fig. 2A, B and Supplementary Table 2). The expansion of $S$ and interlayer spacing contributes to the increase in cell volume. While the change in interlayer spacing is commonly observed in 2D MOFs, the change in the 2D layer itself is rather exceptional[38]. Notably, as the area $S$ expands, the thickness of the layer ($d_3$) decreases by 2.6%, which is similar to the principles of origami mechanics, where overlaid molecular structures at 100 K and 380 K aid in understanding the molecular movement (Fig. 2C, D and Supplementary Fig. 14). The 2D layer exhibits negative thermal expansion (NTE) as the thickness shrinks. The NTE of the thickness influences the transition of the cell volume, but the overall cell volume

increases due to a larger expansion of the interlayer spacing between the 2D layers. The NTE property of the 2D layer in MOFs is significantly rare because most flexible 2D MOFs experience transformation in the interlayer[47]. Also, the two graphs show a little non-linear shape because of solvent effects[48]. Furthermore, when the thermal expansion coefficients ($\alpha$) of PPF-301 were calculated using *PASCal*[49], the colossal thermal expansion was observed along the principal orthogonal axis $X_3$ [$\alpha_{X3} = 170(3)$ M K$^{-1}$] (Supplementary Table 3 and Fig. 15), where X3 is approximately parallel to the $[\bar{1}10]$ crystal axis, responsible for the expansion of the interlayer spacing and area S. The thermal expansion coefficient value of PPF-301 is significantly higher than that of many 2D MOFs (Supplementary Table 4).

The DCS origami pattern found in PPF-301 somewhat deviates from the regular DCS pattern, consisting of square tiles (see Supplementary Fig. 16 and Movie 1). The layer in PPF-301 has four different types of tiles, labeled as A, B, C, and D with colors such as yellow, blue, gray, and dark gray, respectively. A and B tiles are occupied either by Zn SBU or TCMOPP linker, respectively (Supplementary Fig. 17). Upon heating, these four types of tiles are rarely expanded (−2.5–0.5%), suggesting that these tiles cannot be in charge of the area expansion.

To pinpoint the actual expansion movement of PPF-301, we now turn our attention to $\theta_1$ and $\theta_2$, defined by the dihedral angles between

**A**

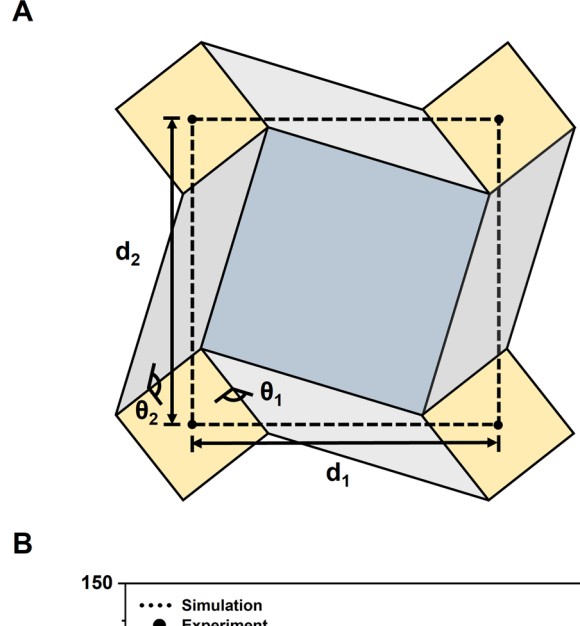

**B**

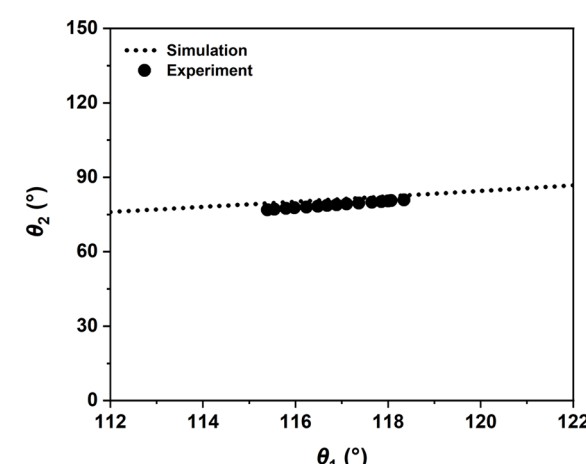

**C**

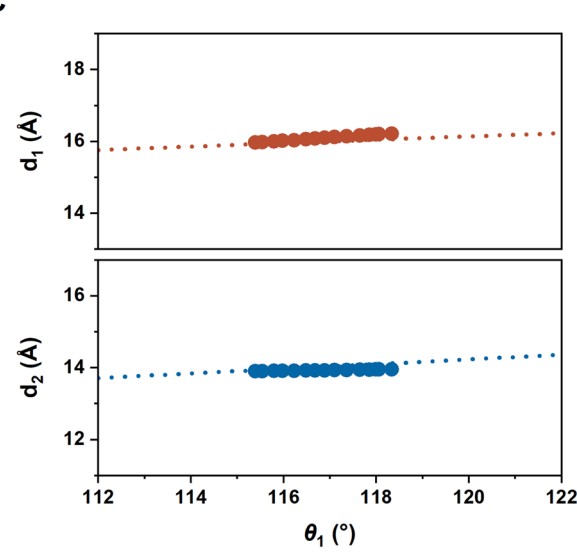

**Fig. 3 | A comparison of an experiment and a mechanical model based on origami tessellation. A** Schematic representation of folding angles $\theta_1$ and $\theta_2$ and lengths $d_1$ and $d_2$. **B** Relationship between the folding angles, $\theta_1$ and $\theta_2$. **C** Relationships between $\theta_1$ and $d_1$ (top) and $\theta_1$ and $d_2$ (bottom). Source data are provided as a Source Data file.

two tiles AC and DA, respectively (see Fig. 3A and Supplementary Fig. 18). According to the crystallographic data, these angles, $\theta_1$ and $\theta_2$, show a steady increase of 2.9° or 3.9°, respectively, from 100 to 380 K (Supplementary Table 5). We compare the experimental data with a geometric model built from a DCS origami tessellation (Fig. 3B and Supplementary Fig. 19A). The relationship between $\theta_1$ and $\theta_2$ is derived from Supplementary Equation (1). As shown in Fig. 3C and Supplementary Fig. 19B, the relationship among $\theta_1$, $d_1$, and $d_2$, is well-matched with the calculated ones from the model, following the equation shown below (see also the definition of $l$, $l'$ and $\alpha$, the relationship between $\theta_1$ and $d_2$ in Supplementary Note 4).

$$d_1 = \sqrt{(l - l' \sin\alpha \cos\theta_1)^2 + l'^2\cos^2\alpha + l'^2\sin^2\alpha\sin^2\theta_1}$$

Therefore, the thermal movement of the 2D layer of PPF-301 is indeed based on the folding mechanism of origami tessellation. In other words, as the 2D layer is flattened, the folding angles ($\theta_1$ and $\theta_2$) increase, just like the origami tessellation (Supplementary Movie 2).

## Origin of origami motion

Thus far, we have identified that the folding movement of the 2D layer was triggered by changing the folding angles ($\theta_1$ and $\theta_2$). To pinpoint the molecular origin of the origami movement resulting from the folding angles, we pay close attention to two dihedral $\varphi_A$ and $\varphi_B$ angles (see Fig. 4A for a definition of $\varphi_A$ and $\varphi_B$), because a transition of the folding angles could be related to these two dihedral angles ($\varphi_A$ and $\varphi_B$) as the nodal point of the framework. When we measured these angles from the crystallographic data, the dihedral angles $\varphi_A$ and $\varphi_B$ changed by 1.0° and 2.4°, respectively (Supplementary Table 6). The variation of $\theta_1$ relies on the linear change of the $\varphi_A$ and $\varphi_B$, as shown in Fig. 4B. We then calculated the potential energy surface (PES) of the isolated aryloxy group while varying these dihedral angles (see Fig. 4C). The dihedral and bond angles from 100 to 380 K are marked on the PES, displaying that the potential well is quite shallow. Searching the Cambridge Structural Database (CSD) reveals that the dihedral angles of most known compounds with aryloxy groups are located within the vicinity of the minimum of our PES. For example, at 100 K, the dihedral and bond angles were populated in a region, where most folded aryloxy groups were found in CSD entries. Interestingly, the molecular structure of TCMOPP in PPF-301 is fairly close to the equilibrium geometry of the isolated aryloxy group. As a result, a flattened layer can be realized without paying a steep penalty in energy. Ultimately, the origami movement is driven by dihedral angle $\varphi$ (C–C–O–C) and bond angle $\alpha$ (C–O–C) of the aryloxy group in the TCMOPP. The inherent flexibility of the TCMOPP linker is the origin of the dynamic movement of the 2D origami framework found in PPF-301.

## Mechanical behavior

To investigate the mechanical properties of PPF-301 based on origami movement, we performed quantum mechanical calculations to construct an optimized structure and calculated total electronic energies using VASP[50] (Supplementary Fig. 20). The average elastic constants were calculated using the ElaStic program (Supplementary Table 7)[51]. Specifically, the maximum and minimum values of the elastic constants were obtained to verify the directional contribution by ELATE software[52]. The spatial dependence of elastic constants is visualized by 3D surfaces and 2D polar plots as shown in Fig. 5A and Supplementary Fig. 21. We found that the softer direction corresponds to the movement between layers along the xy plane. The linear compressibility ($\beta$) and Poisson's ratio ($\nu$) exhibit exceptionally negative values that direction corresponded to the transition of the 2D sheet. The maximum and minimum elastic constants and their directions are represented as shown in Table 1. Young's moduli ($E_{max}$ = 20.05 GPa and $E_{min}$ = 3.79 GPa) of PPF-301 are similar to those of MOF-5[53]. PPF-301 is

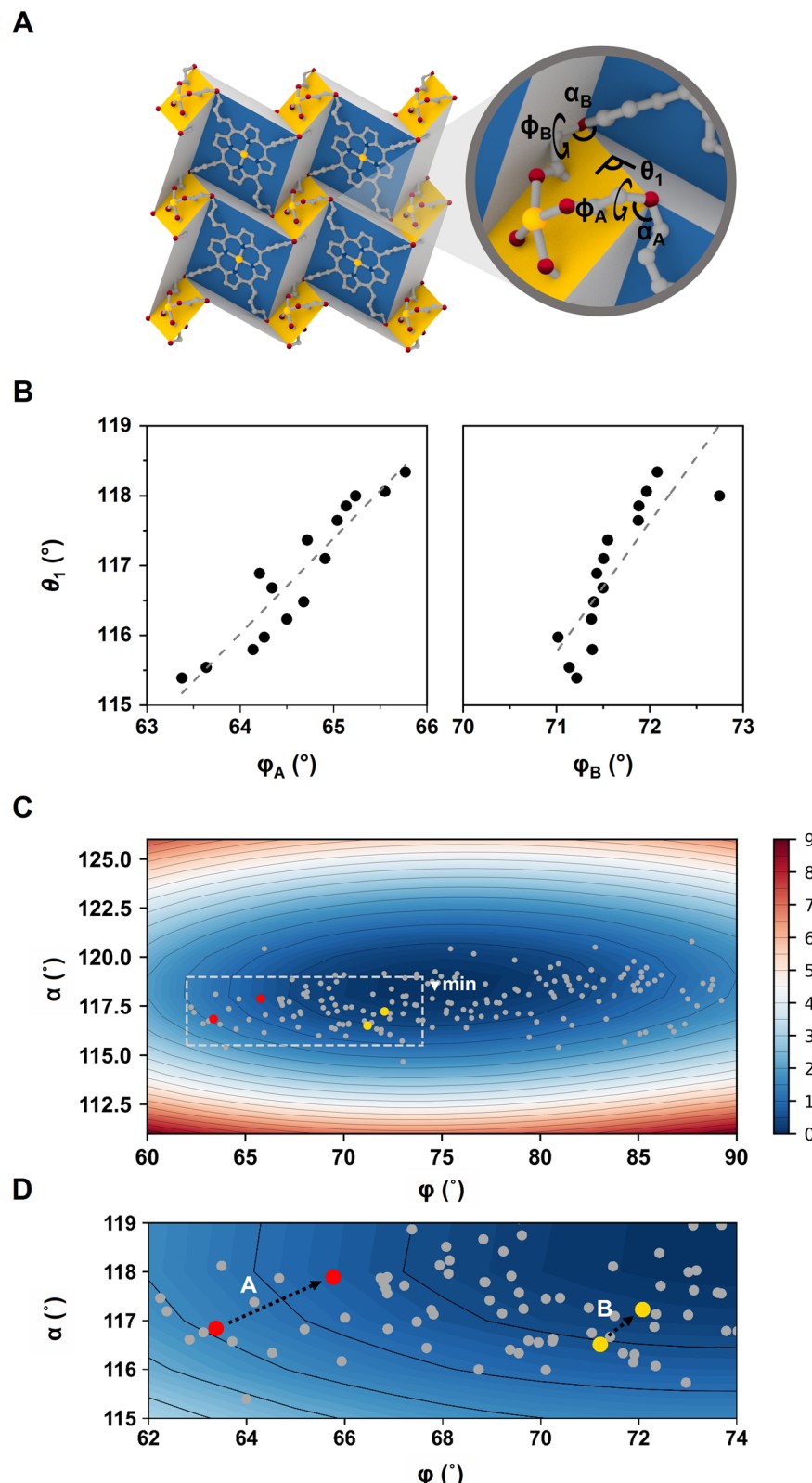

**Fig. 4 | The origin of origami movement in PPF-301. A** Definition of folding angle $\theta_L$, dihedral angle $\varphi$ (C–C–O–C), and bond angle $\alpha$ (C–O–C). There are two types of A ($\varphi_A$, $\alpha_A$) and B ($\varphi_B$, $\alpha_B$) pairs in the framework. Zn = yellow; C = gray; N = blue; O = red; all hydrogen atoms and solvent molecules are omitted for clarity. **B** $\theta$ as a function of $\varphi$. **C** Potential energy surface by varying $\varphi$ and $\alpha$ of the isolated aryloxy group. Gray circles indicate the isolated aryloxy group found in the CSD. **D** The enlarged figure of the dashed box in Fig. 4. As the temperature increased, the two types of angles increased and became energetically stable. Source data are provided as a Source Data file.

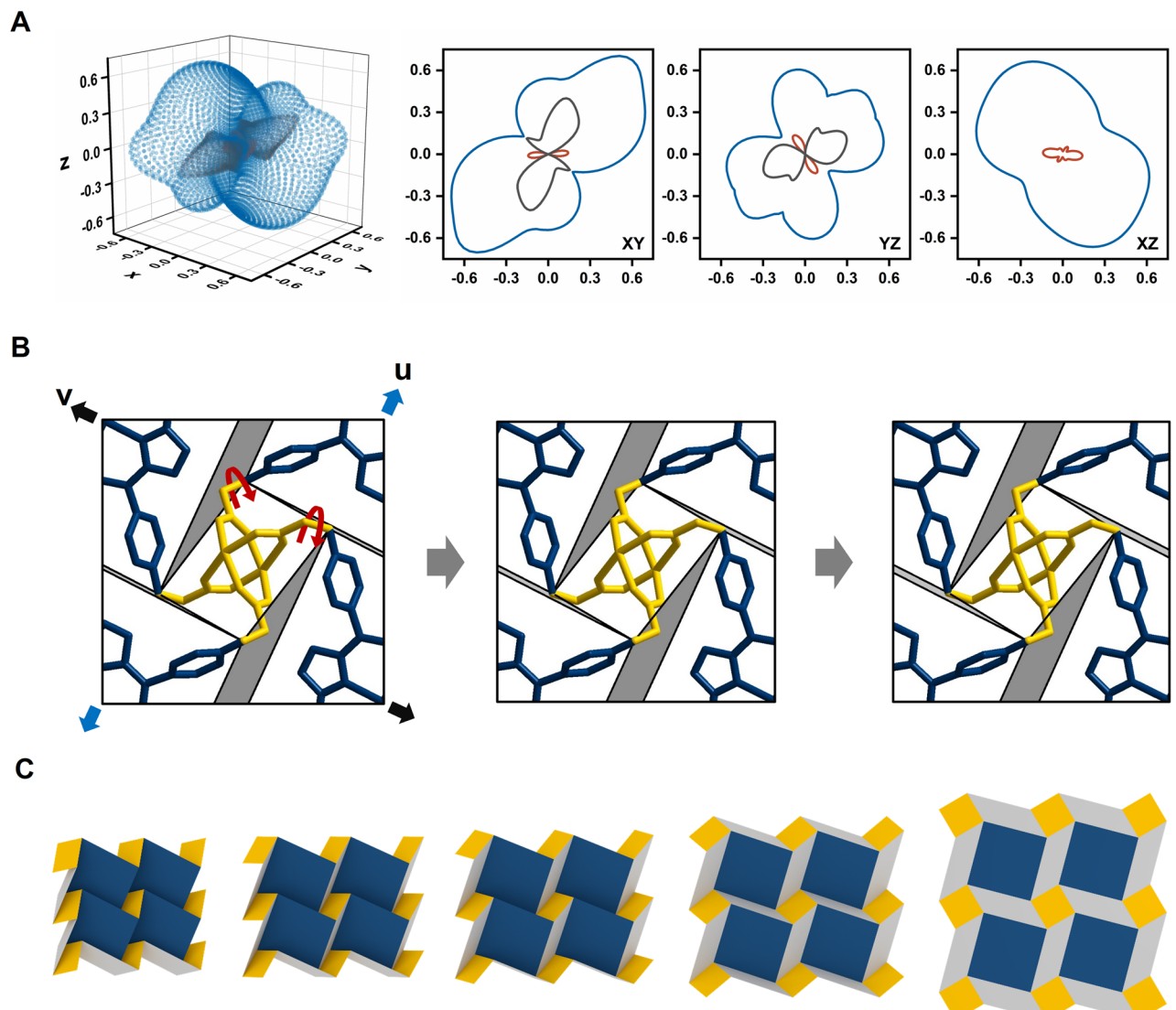

**Fig. 5 | Origami mechanics of PPF-301. A** 3D surfaces and 2D polar plots of Poisson's ratio obtained by ELATE visualization. Blue and black lines represent the maximal and minimal positive values, respectively. The red line represents the minimal negative values over all possible values. **B** Top view of the atomic movement corresponding to the minimal Poisson's ratio. The folded gray areas unfold as the stress is applied along **u** direction, as shown in the figure from left to right in the figure along the gray arrows. blue arrow; **u** = (−0.766, 0.438, 0.471) and black arrow; **v** = (−0.314, 0.385, −0.868) directions. **C** Deployable mechanism of DCS origami tessellation. Source data are provided as a Source Data file.

**Table 1 | The maximum and minimum values of elastic constants**

| Elastic modulus | Young's modulus (GPa) | | Linear compressibility (TPa$^{-1}$) | | Shear modulus (GPa) | | Poisson's ratio | |
|---|---|---|---|---|---|---|---|---|
| Value | $E_{min}$ | $E_{max}$ | $\beta_{min}$ | $\beta_{max}$ | $G_{min}$ | $G_{max}$ | $v_{min}$ | $v_{max}$ |
| | 3.79 | 20.05 | −1.90 | 128.11 | 1.50 | 6.40 | −0.107 | 0.848 |
| Anisotropy (A) | 5.30 | | ∞ | | 4.27 | | ∞ | |
| Axis (**u**) | −0.528 | 0.746 | 0.771 | −0.636 | −0.742 | −0.469 | −0.766 | 0.808 |
| | 0.836 | 0.496 | 0.628 | 0.755 | 0.482 | 0.135 | 0.438 | 0.566 |
| | −0.148 | 0.444 | −0.107 | −0.161 | 0.467 | −0.873 | 0.471 | −0.164 |
| Second axis (**v**) | | | | | −0.113 | −0.599 | −0.314 | −0.589 |
| | | | | | −0.776 | −0.775 | 0.385 | 0.766 |
| | | | | | 0.621 | 0.202 | −0.868 | −0.259 |

slightly anisotropic ($A_E = E_{max}/E_{min}$ as 5.30), when compared to other highly anisotropic materials such as MIL-53(Al)-lp ($A_E = 105$)[54]. Especially, a negative Poisson's ratio (NPR) of PPF-301 is −0.107 along **u** (−0.766, 0.438, 0.471) and **v** = (−0.314, 0.385, −0.868) directions. To investigate the NPR property of the 2D sheet, the change in atomic configurations and cell distortions were analyzed along the planes by applying axial strain along **u**, thereby displaying NPR behavior. When mechanical stress is applied along the **u** direction, the blue and yellow skeletons rotate in opposite directions, inducing the spread out of the folded gray-colored area (Fig. 5B). From the molecular point of view, such motion accompanies the change in dihedral angles and bond angles in the aryloxy group of the linkers.

For the last two decades, several flexible MOFs exhibit abnormal properties such as negative linear compressibility and NPR. However, 2D flexible MOFs, especially, intermolecular 2D layers, are difficult to generate abnormal properties if the 2D layer hasn't had some patterning[38,47,55]. The 2D layer in PPF-301 shows an NPR property, resulting from origami movement through theoretical and experimental investigations. The origami-inspired materials exhibit a wide range of NPR, as origami patterns vary[22]. We also establish a structural model of a 2D corrugated framework, manifesting a deployable mechanism based on the folding−unfolding motion (see Fig. 5C). In summary, PPF-301 with a DCS origami tessellation shows NPR behavior as an origami metamaterial[19,22].

## Discussion

The discovery of dynamic crystals completely changed the general idea of solids that were considered a "chemical cemetery"[56,57]. Especially, flexible MOFs have exhibited astonishing transformation based on abundant molecular building blocks, organic linkers and metal nodes. Local movements of these building blocks, such as bending[58], twisting[59], and rotating[60] triggered dynamic behavior, swelling[61], and breathing[62]. Such dynamic behavior of MOFs can be predicted through topological analysis[37]. Significantly, the hidden dynamic behavior of MOFs is unveiled by the usage of flexible geometries instead of the existing topologies, as exemplified by meta-MOF, UPF-1, as square tessellation, identified in mathematics[43]. The structural analysis ultimately leads to the discovery of folding behavior driven by structural flexibility.

We report an origamic MOF, PPF-301, assembled from flexible porphyrin linkers and Zn SBUs. A 2D porphyrinic sheet of PPF-301 shows the folding movement based on a DCS origami tessellation at the molecular level. We demonstrate that the folding movement of origami mechanics originated from a change of the dihedral and bond angles in the aryloxy group of the flexible linker, as pivot points. Generally, in 2D MOFs, solvents play a crucial role in maintaining the packing between 2D layers. Different types of solvents can affect the packing arrangement of these layers[33,63]. In the case of the PPF-301 structure, the degree of folding in the 2D layer varies depending on the solvents used, in addition to the stacking pattern. Interestingly, the origami tessellation of the 2D layer, observed in Fourier-filtered images of HR-TEM, remains preserved regardless of the solvent (Fig. 6A−C). We observe that the inherent crumpled pattern of the 2D layer is maintained, suggesting that the folding mechanism remains the same regardless of the solvents, as the DCS pattern allows for one degree of freedom in deformation (Fig. 6D−F). Further research could be proposed by creating precisely controlled nanosheets to observe a more considerable movement of the 2D layer itself, to confirm the effect of the solvent and crystal size[64].

Notably, origami tessellations, closely related to flexible geometry, provide the development of origamic MOFs, opening a distinct category of MOF metamaterials with mechanical properties[42]. Also, the origami tessellations can be applied to advanced design principles of MOFs to assemble dynamic frameworks exhibiting origamic movements. Otherwise, to alter the flexibility of the 2D layer itself for origamic MOFs, we can explore the incorporation of diverse functional groups such as -CH₂-, -S-, and -NH- instead of the aryloxy group. The preferred dihedral and bond angles associated with each functional group act as pivot points, leading to varying degrees of folding movement (Supplementary Fig. 22). Furthermore, the folding movement based on origami tessellation shown here allows controlling the distance between metal nodes upon external stimuli, which could potentially provide 2D spin qubit frameworks[65,66] to develop advanced molecular quantum computing− one of the future applications remaining for origamic MOFs.

## Methods

### Synchrotron powder X-ray diffraction (PXRD)

X-ray powder diffraction data were collected at the 2D SMC, PAL beamline (2023-2nd-2D-030) at the Pohang Accelerator Laboratory (PAL) in the Republic of Korea. The as-synthesized PPF-301 crystals were finely ground under wet conditions. The prepared powders were then packed and sealed into a capillary with a diameter of 0.3 mm (wall thickness, 0.01 mm). The PXRD data were collected at 298 K with a Rayonix MX225HS CCD detector.

### Temperature-dependent synchrotron single-crystal X-ray diffraction (SCXRD)

A crystal was sealed in a capillary having 0.3 mm in diameter (wall thickness: 0.01 mm) and a small amount of mother liquid was filled in the capillary to generate vapor. Its single-crystal X-ray diffraction was collected at 2D SMC, PAL with synchrotron light source ($\lambda = 0.63000$ Å) and Si(111) double crystal monochromator. Rayonix MX225HS CCD area detector was used at a 66.00 mm distance. Temperature-dependent SCXRD data were collected from 100 to 380 K at intervals of 20 K. It is controlled from PAL BL2D-SMDC program[67] using the Cryojet 5 system and stabilized for 15−20 min. Data processing such as cell refinement, reduction, and absorption correction was performed using HKL3000 (Ver. 720)[68]. The crystal structures of PPF-301 were solved by the intrinsic phasing method and refined by full-matrix least-squares calculations with the SHELXL program[69]. The final refinement was performed with the modification of the structure factors for the electron densities of the disordered solvents using the SQUEEZE option of PLATON[70]. The crystallographic data for PPF-301 depending on temperatures was deposited in the Cambridge Crystallographic Data Centre (CCDC 2122043−2122057).

### Thermal expansion coefficient ($\alpha$)

The thermal expansion coefficient of PPF-301 was calculated based on SCXRD data by the PASCal software[49].

### Thermogravimetric analysis (TGA)

TGA was conducted using a TA instrument SDT Q600, with heating performed from 303 K to 1073 K under an $N_2$ atmosphere at a scan rate of 10 K min⁻¹.

### Gas sorption

A gas sorption study was performed on a Micromeritics ASAP 2020 instrument.

### ¹H Nuclear magnetic resonance (NMR) spectroscopic analysis

¹H NMR data were collected on Agilent FT-NMR (400-MR DD2) spectrometer.

### Elemental analysis (EA)

The EA experiment was conducted using ThermoFisher Scientific Flash 2000 at the UNIST Central Research Facilities Center.

### Fourier-transform infrared (IR) spectroscopic analysis

IR spectra were recorded using a ThermoFisher Scientific Nicolet iS10 FT-IR spectrometer equipped with an ATR detector.

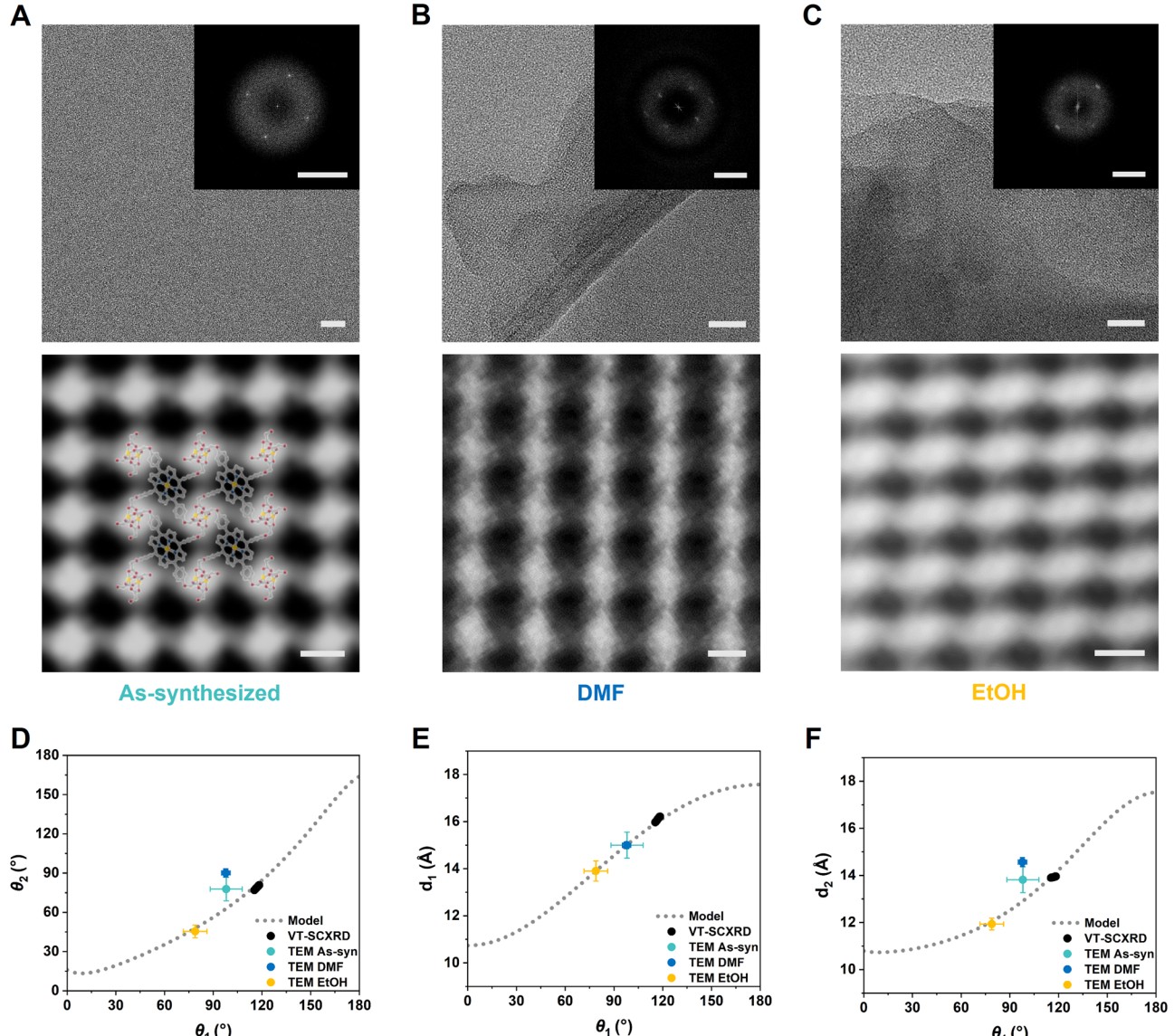

**Fig. 6 | Origami tessellation analysis of samples exchanged with different solvents.** HR-TEM images of (**A**) as-synthesized, (**B**) DMF, and (**C**) EtOH samples (scale bar: 20 nm), the corresponding fast Fourier-transform (FFT) patterns are inserted (scale bar: 1 nm$^{-1}$) in the upper figure. Fourier-filtered images with the PPF-301 structure superimposed (scale bar: 1 nm) in the lower figure. **D** The relationships between $\theta_1$ and $\theta_2$, as well as (**E**, **F**) the relationships among $d_1$ and $d_2$ and $\theta_1$ along the dotted line obtained from the mathematical model, were examined. The average values of $d_1$ and $d_2$, measured from Fourier-filtered images of each sample, are marked on all of the graphs. $\theta_1$ and $\theta_2$ were calculated using Supplementary Equations (1) and (2). Source data are provided as a Source Data file.

## High-resolution transmission electron microscopy (HR-TEM) analysis

Before measuring the HR-TEM, the as-synthesized PPF-301 crystals were prepared either by washing them in the mother liquid, or by solvent-exchanging them with DMF and EtOH for 1 mL × 3 times, respectively. The prepared crystals were subsequently crushed through ultrasonication for 30 min. After ultrasonication, a droplet of the resulting suspension was transferred onto a carbon-film copper grid and allowed to dry for 12 h in preparation for HR-TEM measurement. A comparison between experimental data obtained from HR-TEM measurements and a mathematical model. HR-TEM analysis was performed with a JEM-2100 microscope (JEOL Company) equipped with a LaB$_6$ electron gun operated at an acceleration voltage of 200 kV. Generating Fast Fourier-Transform patterns and image filtration were conducted by using Gatan DigitalMicrograph software.

## Synthesis of PPF-301

2D porphyrinic MOF, PPF-301, was synthesized by a slightly modified procedure[45]. TCMOPP (4.5 mg, 0.005 mmol) and Zn(NO$_3$)$_2$·6H$_2$O (4.1 mg, 0.014 mmol) were added to a solution of DMF/EtOH (1.0 mL, 3:1) in 16 mL vial. 1 N HNO$_3$ (15 μL) was added to the solution. The mixture was sealed and sonicated to assure homogeneity. After then, the solution was heated at 80 °C for 24 h, followed by slow cooling to room temperature for 9 hours yielding purple crystals. The obtained crystals were washed with mother liquid (3 × 3 mL), filtered, and collected. Anal. Cal. for Zn$_3$C$_{70}$H$_{70}$N$_{10}$O$_{18}$ [Zn$_3$(TCMOPP)(DMF)$_6$]; C, 54.60; H, 4.85; N, 9.10. Found. C, 53.4 ± 0.1; H, 4.17 ± 0.05; N, 8.41 ± 0.01.

## Reporting summary

Further information on research design is available in the Nature Portfolio Reporting Summary linked to this article.

## Data availability

All of the data generated in this study are provided in the Supplementary Information/Source Data file. Crystallographic data for PPF-301 structures (from 100 to 380 K at intervals of 20 K), which depend on variable temperatures used in this study, have been provided in Supplementary Data 1 and deposited at the Cambridge Crystallographic Data Centre under deposition numbers CCDC 2122043–2122057 [https://www.ccdc.cam.ac.uk/structures/]. Source data are provided with this paper.

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

## Acknowledgements
This work was supported by the National Research Foundation (NRF) of Korea (NRF-2021M3I3A1084909, NRF-2020R1A2C3008226, NRF-2021R1A3B1077184, and NRF-2016R1A5A1009405) and Korea Environment Industry & Technology Institute (KEITI) through Public Technology Program based on Environmental Policy Program, funded by Korea Ministry of Environment (MOE) (2018000210002). E.J. acknowledges the Global PhD Fellowship (NRF-2017H1A2A1042129). I.S.L. and S.K.M. used the supercomputer Aleph, supported by the IBS Research Solution Center. The single-crystal structure was collected at BL2D SMC (2020-2nd-2D-M005 and 20230-2nd-2D-030), Pohang Accelerator Laboratory. We would like to express our gratitude for the support and advice provided by Dr. Tobias Ritschel and Dr. Alexander Mistonov from Prof. Dr. Jochen Geck's group in the Institute of Solid State and Materials Physics, as well as Dr. Volodymyr Bon from Prof. Dr. Stefan Kaskel's group in the Department of Inorganic Chemistry at Technische Universität Dresden, for their assistance in the pressure-induced single crystal X-ray diffraction experiment.

## Author contributions
Conceptualization, E.J. and W.C.; Methodology, E.J., H.C., and W.C.; Crystal structural analysis, E.J. and D.M.; Simulation, I.S.L., S.K.M., and D.C.Y.; Structural visualization, E.J. and E.K.; Structural modeling, E.J. and J.N., Linker synthesis, E.J., J.L., and H.-J.N.; Writing—Review & Editing, E.J., I.S.L., D.C.Y., S.K.M., and W.C.

## Competing interests
The authors declare no competing interests.
