## [Peer Review File · Nature Communications]

Origamic Metal-Organic Framework toward Mechanical MetamaterialREVIEWER COMMENTS

Reviewer #1 (Remarks to the Author):

The authors describe a fascinating, novel aspect of 2D MOFs behaving as “soft crystalline solid-state materials” and showing structural flexibility which mimics (macroscopic) origami folding mechanisms. The discovery is based on tetratopic porphyrine-linkers (i.e., 5,10,15,20-tetrakis [4-(72 carboxymethyleneoxyphenyl)] porphyrin, TCMOPP), functionalized by choosing conformationally flexible ligand groups (carboxymethyleneoxyphenyl), that are connected via tetratopic Zn paddle wheel SBUs. The work is conceptually situated in exploiting flexible MOFs for various applications (i.e. stimuli responsive (meta-)materials and in particular it builds upon previous work of the authors and others on flexible porphyrine tetracarboxylic acids for crystal engineering (e.g. refs 41, 42, 43). The conceptual novelty and originality of the work is well argued in the manuscript at several sections. Systematically exploiting 2-D MOFs as a materials platform for reversible structural transformations mimicking origami folding is new.

The work has largely been conducted with care and at the state of art (few technical comments see below). It is mainly based on detailed analysis of temperature dependent single crystal X-ray diffraction data (cell volume, interlayer spacing, dihedral angles etc.) and rationalization of these “movements” by a mathematical model for the DCS origami pattern and the respective folding mechanism (“We compare the experimental data with a geometric model built from a DCS origami tessellation”). This analysis and discussion of the data has been convincingly documented (MS and SI). The observed “colossal” thermal expansion of PPF-301 as a result of the folding mechanism is impressive. The origin of the specific motion is convincingly analyzed and mechanistically ascribed to the PES of the aryloxy groups. PPF-301 shows negative linear compressibility and NPR. Thus, the key point of the work is demonstrating a rational concept of how to integrate origami folding / unfolding mechanisms to a very versatile materials class (2-D MOFs).

Nevertheless, the work may still have a weak point in terms of fundamental understanding of the underlying phenomena, as it is limited to the “mechanical model” of the movement. A comprehensive thermodynamic analysis of the flexible behavior of PPF-301 (temperature and pressure dependent SCXRD) and the disentanglement of enthalpic and entropic driving forces including the role of solvate (of other guest) molecules of the responsive behavior is clearly important. Also, a number of potential areas of application or impact of the new concept of 2D-MOF metamaterial design are mentioned in the conclusion, however the work itself does not present a convincing step in one of these directions.

There are few technical issues which the authors may consider:

- a) A full data set of chemical analysis of PPF-301 should be given, proving the chemical identity, sample purity etc. including elemental analysis (C,H,N, Zn), spectroscopic features (e.g. IR), quantification of included solvent molecules etc.
- b) Why does the as-synthesized PXRD pattern (SI Fig. 1) deviate quite substantially from the calculated

one based on the SCXRD data?

c) The structure determination via SCXRD is described; however, a more comprehensive structure description should be given. SI Fig.2 does only focus on the packing mode of the folded 2D-MOF sheets – it does not really specify the details of the MOF structure itself. Implicitly, and evident for the MOF “expert” as a reader, the things are clear (Zn-metallated porphyrine, arrangement of the Zn paddlewheels, etc.) However, a more elaborate and labeled presentation of the structural aspects should be provided.

d) The study of the folding/unfolding mechanism is done with the solvated PPF-301. “A crystal was sealed in a capillary having 0.3 mm in diameter (wall thickness: 0.01 mm) and a small amount of mother liquid was filled in the capillary to generate vapor.” The authors do not tell anything about the role of the included solvent molecules. Is there a dependence on the motions by the choice of solvents or guests? What happens if the PPF-301 is desolvated (is that possible without structural disintegration?). If the mechanism were dependent on the solvated structure, and would disappear in the desolvated material how can the authors claim a very broad and fundamental impact of their discovery? Would this dependence limit the potential applicability of PPF-301?

In general, I am in favor of this manuscript, congratulate the authors to their discovery, and would recommend acceptance if the authors convincingly deal with the issues raised.

Reviewer #2 (Remarks to the Author):

In the work titled “Foldable Metal-Organic Framework as Origamic Mechanical Metamaterial”, Jin et al. report a kind of two-dimensional porphyrinic MOF which could display folding motions based on origami tessellation. By performing DFT calculations, the authors reveal the folding mechanism of a 2D porphyrinic sheet PPF-301 as changes in the dihedral and bond angles in the aryloxy group of the flexible liner. Even though the proposed idea of origami mechanical metamaterials is fascinating, however, most of the data were based on theoretical calculation, and no direct evidence (e.g., SEM or TEM photos) shows the flexible folding movement actually occurred on the 2D MOFs. After careful review and consideration, we believe this manuscript is more suitable for a specialized journal about computational chemistry instead of Nature Communications. Below are some detailed comments:

1. The authors should provide more experimental data to illustrate the proposed origamic flexible folding movement behavior of the 2D MOFs, such as scanning electron microscopy (SEM) or transmission electron microscopy (TEM) images directly showing the morphology changes.
2. The authors propose the 2D MOFs as mechanical metamaterials, however, no experimental results about the mechanical properties of the MOFs were found throughout the manuscript (including in the SI). X-ray diffraction and thermal analysis results are hard to reflect the intrinsic mechanical properties of MOFs, which makes it difficult to correlate the simulation results to the actual behavior of the materials.
3. In the manuscript, according to the DFT calculation, how can origami tessellation improve the properties of MOF to make it a mechanical metamaterial? What is the relationship between the folding angles and mechanical properties of MOF?
4. Is DCS origami tessellation applicable to most two-dimensional MOFs, and what are the applicable conditions?

Reviewer #3 (Remarks to the Author):

This paper describes a mechanistic insight into the flexibility of a two-dimensional metal-organic framework (MOF). The authors found one reported two-dimensional MOF, here named PFF-301 (Goldberg et al. CrystEngComm 2010, 4095.), assembled from zinc ions and porphyrin ligand (TCMOPP), shows Origami-like tessellation in response to temperature. This “paper folding” behavior of two-dimensional MOF was further structurally correlated to the specific angle of chemical bonds. Indeed, this paper is similar to the work done by the same authors (Metal-organic framework based on hinged cube tessellation as transformable mechanical metamaterial, Sci. Adv. 2019, eaav4119). The previous Sci Adv paper discussed similar tessellation behavior based on three-dimensional MOF. Even though I see a certain novelty in this paper such as 2D MOF and paper-folding tessellation mechanism), what the authors demonstrated experimentally, such as temperature-dependent single crystal X-ray diffraction experiments, looks routine. I would happily accept this paper if the authors could have shown another stimulus to induce the structural change. At this moment, I am not so convinced that this paper showed the novelty criterion for publication in Nature Communications. The more detailed scientific concerns and discussions are as follows.

1. Both this and previous papers discussed the mechanical metamaterial behavior of MOF only by structural change in response to temperature. Of course, the positive or negative thermal expansion of the framework should be first investigated; however, I was expecting that another stimulus could change the structure of this system because the authors discovered such metamaterial-like behavior already in the previous paper. For instance, pressure-dependent SCXRD measurement could have been carried out to see the real effect of mechanical stress on structural change.
2. Another way to improve this paper would be generalizability. By understanding this mechanism, the authors might be able to suggest any other molecular system that can show this origami tessellation. If the authors are able to design such a new ligand or metal node and demonstrate it, this paper will become more robust and more appealing.
3. After showing the correlation between the molecular structure and the simplified schematic model in Figure 2, there is no molecular model in the figure. This makes it more difficult to understand how much the structure changed in response to temperature. It would be better to show and overlay two extreme cases of origami tessellation for clarity.

Reviewer #4 (Remarks to the Author):

This manuscript proposes a new idea to use foldable MOF as origamic architected metamaterial, which provides a new approach for designing and building mechanical materials at molecular level, with adjustable properties. The concept is novel and interesting, but the execution seems not comprehensive and fully convincing, despite that the authors conducted a thorough experimental investigation of the

MOF mechanical properties complemented with computational modelling to demonstrate the rational design of various MOF-based architectures. The experimental part (including manufacturing and characterization) and actual demonstration should be substantially enhanced.

Some specific comments to be addressed as well:

1. Based on the literature, there are some general work principles and applications of metal-organic frameworks (MOFs) at the molecular level. However, in the intro section, the author describes the origami structure in detail (well done), which is not however very friendly to those who are unfamiliar with MOFs. What are the uniqueness and advantages of the proposed approach compared with previous work? The author did mention parts of that in lines 56-62. Nevertheless, it is not clear to me. The author may need to state them clearly.

2. The author aim to implement a design approach of origamic mechanical metamaterial, but the optimization analysis of the structure towards some functionality, e.g., reconfigurability of the structure, or specific stiffness beyond ensuring the foldability of the origami structure is missing. At the moment, the authors discuss the negative thermal expansion (NTE) of 2D layer, but the discussion of functionality of the original structure is lost (i.e., foldability). What is specifically the objective function in their approach, and what are the constraints?

3. The authors did not discuss the scalability of these MOF-based metamaterials, the feasibility of metamaterials under subsidy parameters is verified in the Supplementary materials, but there is no further study of the versatility of other origami structures and materials, and there is no in-depth discussion of the manufacturability, material cost, and time to produce these metamaterials, which are important (otherwise would be just some MOF folding simulation)

4. While the article briefly mentions several potential applications for MOF-based metamaterials, such as flexible and mobile electronics, self-adaptive sensor systems, and soft robotics. However, there was no discussion regarding the practicality, feasibility, and challenges of utilizing these metamaterials in real-world applications, especially on what specific applications this proposed origamic mechanical metamaterial would be best for, and why the specific capabilities of the MOF are critical to enable such applications.

Some minor issues:

1. The temperature unit should be uniform. Line 79 is Celsius; the rest is Kelvin.
2. As an original research, the Fig. 1 appears not very necessary...

Reviewer comments

Reviewer #1 (Remarks to the Author):

The authors describe a fascinating, novel aspect of 2D MOFs behaving as “soft crystalline solid-state materials” and showing structural flexibility which mimics (macroscopic) origami folding mechanisms. The discovery is based on tetratopic porphyrine-linkers (i.e., 5,10,15,20-tetrakis [4-72 carboxymethyleneoxyphenyl] porphyrin, TCMOPP), functionalized by choosing conformationally flexible ligator groups (carboxymethyleneoxyphenyl), that are connected via tetratopic Zn paddle wheel SBUs. The work is conceptually situated in exploiting flexible MOFs for various applications (i.e. stimuli responsive (meta-)materials and in particular it builds upon previous work of the authors and others on flexible porphyrine tetracarboxylic acids for crystal engineering (e.g. refs 41, 42, 43). The conceptual novelty and originality of the work is well argued in the manuscript at several sections. Systematically exploiting 2-D MOFs as a materials platform for reversible structural transformations mimicking origami folding is new.

The work has largely been conducted with care and at the state of art (few technical comments see below). It is mainly based on detailed analysis of temperature dependent single crystal X-ray diffraction data (cell volume, interlayer spacing, dihedral angles etc.) and rationalization of these “movements” by a mathematical model for the DCS origami pattern and the respective folding mechanism (“We compare the experimental data with a geometric model built from a DCS origami tessellation”). This analysis and discussion of the data has been convincingly documented (MS and SI). The observed “colossal” thermal expansion of PPF-301 as a result of the folding mechanism is impressive. The origin of the specific motion is convincingly analyzed and mechanistically ascribed to the PES of the aryloxy groups. PPF-301 shows negative linear compressibility and NPR. Thus, the key point of the work is demonstrating a rational concept of how to integrate origami folding / unfolding mechanisms to a very versatile materials class (2-D MOFs).

Nevertheless, the work may still have a weak point in terms of fundamental understanding of the underlying phenomena, as it is limited to the “mechanical model” of the movement.

A comprehensive thermodynamic analysis of the flexible behavior of PPF-301 (temperature and pressure dependent SCXRD) and the disentanglement of enthalpic and entropic driving forces including the role of solvate (of other guest) molecules of the responsive behavior is clearly important.

Also, a number of potential areas of application or impact of the new concept of 2D-MOF metamaterial design are mentioned in the conclusion, however the work itself does not present a convincing step in one of these directions.

We would like to thank the reviewers for their constructive input on this manuscript. In this revision, we include transmission electron microscopy (TEM) measurement and pressure-induced single-crystal X-ray diffraction (SCXRD, see details in A4, B2) to strengthen fundamental understanding of the underlying folding mechanism of the title 2D MOF, PPF-301. We also elucidate the role of the solvent in PPF-301 (see A4–6). Additionally, we propose a designable 2D metamaterial based on the calculated potential energy surface of the functional group and have revised the description of the specific applications (see C2 and D5).

There are few technical issues which the authors may consider:

(1) a) *A full data set of chemical analysis of PPF-301 should be given, proving the chemical identity, sample purity etc. including elemental analysis (C,H,N, Zn), spectroscopic features (e.g. IR), quantification of included solvent molecules etc.*

A-1. We have included a complete dataset of chemical analysis for PPF-301 which includes elemental analysis (C, H, N), Fourier-transform infrared (FT-IR) spectroscopic analysis, ^1H Nuclear magnetic resonance (NMR) spectroscopic analysis, and optical microscopic images.

Elemental analysis: The elemental analysis data has been added as the following on page 14, lines 9–10 and page 15, lines 1–3 of the “Materials and Methods” section in the manuscript.

“**Elemental analysis (EA).** The EA experiment was conducted using ThermoFisher Scientific Flash 2000 at the UNIST Central Research Facilities Center.”

“The obtained crystals were washed with the mother liquid (3×3 mL), filtered, and collected. Anal. Cal. for $\text{Zn}_3\text{C}_{70}\text{H}_{70}\text{N}_{10}\text{O}_{18}$ [$\text{Zn}_3(\text{TCMOPP})(\text{DMF})_6$]; C, 54.60; H, 4.85; N, 9.10. Found. C, 53.4 ± 0.1 ; H, 4.17 ± 0.05 ; N, 8.41 ± 0.01 .”

Fourier-transform infrared (FT-IR) spectroscopic analysis: FT-IR data has been added as follows on page 5, lines 20–21 and page 14, lines 11–12 of the “Materials and Methods” section in the manuscript, as well as page 12 of the supplementary information.

“The coordinated DMF molecules were observed using FT-IR spectra in addition to crystallographic data (Supplementary Figure 6).”

“**Fourier-transform infrared (FT-IR) spectroscopic analysis.** IR spectra were recorded using a ThermoFisher Scientific Nicolet iS10 FT-IR spectrometer equipped with an ATR detector.”

Supplementary Figure 6. FT-IR spectra of TCMOPP and PPF-301. The $\nu(\text{C=O})$ and $\delta(\text{O=C-N})$ bands indicate the presence of coordinated DMF molecules in PPF-301.

^1H Nuclear magnetic resonance (NMR) spectroscopic analysis: ^1H NMR experiment was conducted to verify the solvent content of PPF-301. ^1H NMR data was obtained from an acid-digested sample.

The crystals contained TCMOPP linker and DMF molecules in a ratio of 1.08:5.04, as shown in Supplementary Figure 7 on page 13 of the supplementary information.

Supplementary Figure 7. ^1H NMR spectrum showing the trace of solvent content in PPF-301. The crystals were washed with the mother liquid, filtered, and then digested using 0.5 mL of DMSO- d_6 and 0.1 mL of dilute DCl (0.1 mL of 35 % DCl in D_2O in 0.5 mL DMSO- d_6). The ratio of TCMOPP to DMF was determined to be 1.08:5.04.

We have added the following information on page 5, lines 21–24 of the manuscript.

“The solvent content of the as-synthesized PPF-301 was confirmed through a ^1H NMR experiment. The ratio of TCMOPP to DMF was found to be 1.08:5.04, which is similar to the ratio obtained from the crystallographic data (Supplementary Fig. 7).”

Optical microscopic image: We observed the as-synthesized PPF-301 crystals using optical microscopy. The crystals exhibited a uniformly observed rectangular plate shape, as depicted in Supplementary Figure 2 (page 8 in the supplementary information).

Supplementary Figure 2. Optical microscope images of PPF-301. The crystal exhibits a rectangular plate shape with a pale purple color (Scale bar: 0.1 mm).

We have added the following information on page 5, lines 4–5 of the manuscript.

“The as-synthesized PPF-301 crystals display a pale purple color and exhibit a rectangular plate shape (Supplementary Fig. 2).”

(2) b) Why does the as-synthesized PXRD pattern (SI Fig. 1) deviate quite substantially from the calculated one based on the SCXRD data?

A-2. The morphology of the PPF-301 crystal is characterized by a rectangular plate shape (see the added optical microscope images in A-1). In the as-synthesized PXRD pattern, the (01-1) and (11-1) peaks exhibit strong intensities. The differences in diffraction intensities can be mainly attributed to the preferred orientation of the (*h*1-1) reflection in the PPF-301 crystal. We have revised the as-synthesized PXRD data, which was obtained at the Pohang Accelerator Laboratory (PAL), to enhance its quality. Please refer to Supplementary Figure 4 (page 10 in the supplementary information) for the updated PXRD data.

Supplementary Figure 4. Synchrotron powder X-ray diffraction data of the as-synthesized PPF-301. The simulated reflection data is compared with the experimental results.

We have revised the following information on page 13, lines 2–7 of the “Materials and Methods” section in the manuscript.

“Synchrotron powder X-ray diffraction (PXRD). X-ray powder diffraction data were collected at the 2D SMC, PAL beamline (2023-2nd-2D-030) at the Pohang Accelerator Laboratory (PAL) in the Republic of Korea. The as-synthesized PPF-301 crystals were finely ground under wet conditions. The prepared powders were then packed and sealed into a capillary with a diameter of 0.3 mm (wall thickness, 0.01 mm). The PXRD data were collected at 298 K using a Rayonix MX225HS CCD detector.”

(3) c) The structure determination via SCXRD is described; however, a more comprehensive structure description should be given. SI Fig.2 does only focus on the packing mode of the folded 2D-MOF sheets – it does not really specify the details of the MOF structure itself. Implicitly, and evident for the MOF “expert” as a reader, the things are clear (Zn-metallated porphyrine, arrangement of the Zn paddlewheels, etc.) However, a more elaborate and labeled presentation of the structural aspects should be provided.

A-3. We have revised the structural description of PPF-301 on page 5, lines 2–20 of the manuscript, and added a specifically labeled figure as shown in Supplementary Figure 3 (page 9 in the supplementary information).

“PPF-301. PPF-301 crystals were synthesized with $\text{Zn}(\text{NO}_3)_2 \cdot 6\text{H}_2\text{O}$ and 5,10,15,20-tetrakis [4-carboxymethyleneoxyphenyl] porphyrin (TCMOPP) (Supplementary Scheme 1) via a solvothermal reaction. The as-synthesized PPF-301 crystals display a pale purple color and exhibit a rectangular plate shape (Supplementary Fig. 2). PPF-301 consists of Zn paddlewheel SBUs and ZnTCMOPP in a 2 to 1 stoichiometric ratio (Fig. 1a), forming 2D layers. During the reaction, the porphyrin core in TCMOPP undergoes metallation, resulting in a five-coordinate Zn ion that coordinated the DMF solvent. The functionalized aryloxy group of the porphyrin backbone allows for two different orientations, forming the Zn SBUs. Two DMF solvents coordinate with the exterior axial position of each Zn SBUs (Supplementary Fig. 3). Notably, the self-assembled 2D layer of PPF-301 exhibits a corrugated structure due to the flexible aryloxy groups in the TCMOPP linker, which is in contrast to another porphyrinic MOF, PPF-1, where 2D square grids are built from a rigid tetratopic porphyrinic linker⁴⁴. The synchrotron powder X-ray diffraction pattern of the as-synthesized PPF-301 matches well with the simulated pattern (Supplementary Fig. 4) and shows an isostructure with a 2D porphyrinic MOF reported by the Goldberg group.⁴⁵ The 2D layers in PPF-301 are alternately stacked in a stepwise fashion with an interlayer distance of 7.8 Å, parallel to $(\bar{1}11)$ plane of the crystal (Supplementary Fig. 5). In this stacking system, free DMF solvents are observed between the 2D layers. The presence of solvents in the framework contributes to the interaction between interlayers, maintaining a close interval.”

Supplementary Figure 3. The asymmetric unit of PPF-301 based on crystallographic data obtained at 100 K. Copper, carbon, nitrogen, and oxygen atoms are indicated in orange, gray, blue, and red, respectively. Hydrogen atoms have been omitted for clarity. (Displacement ellipsoids are shown at a 50 % probability).

(4) d) *The study of the folding/unfolding mechanism is done with the solvated PPF-301. “A crystal was sealed in a capillary having 0.3 mm in diameter (wall thickness: 0.01 mm) and a small amount of mother liquid was filled in the capillary to generate vapor.” The authors do not tell anything about the role of the included solvent molecules. Is there a dependence on the motions by the choice of solvents or guests?*

A-4. PPF-301 consists of stacked 2D layers composed of Zn paddlewheel SBUs, which exhibit low structural stability without solvents. Examples are the stacked arrangement of the 2D layers, maintained by the presence of solvents i.e. see *Angew. Chem. Int. Ed.* **47**, 8843–8847 (2008), *Chem.*

Rev. **121**, 3751–3891 (2021). Therefore, to perform temperature-dependent SCXRD measurements over an extended period of time (at least 8 hours per sample), we prepared a sample that was sealed in a capillary with a small amount of solvent to preserve the structure while ensuring the integrity of crystallinity.

We have included the following information on page 6, lines 16–18 of the manuscript.

“During the experiment conducted over an extended period of time, we prepared a crystal in a sealed capillary, including a small amount of solvent to prevent any loss of crystallinity.”

To investigate whether the motion of the 2D layer is independent of the solvent, we conducted a solvent exchange experiment from the mother liquid to three different solvents: dimethyl sulfoxide (DMSO), DMF, and EtOH. In the OM images (scale bar: 0.1 mm), we observed cracks in the crystals although the morphology remained intact. The PXRD patterns of the samples exchanged with DMSO and EtOH exhibited changes, including a decrease in intensity, unlike the sample exchanged with DMF. The discrepancy in PXRD patterns between the as-synthesized sample and solvent-exchanged samples can be attributed to the packing system of the 2D layers. The solvents that are incorporated between the 2D layers play a critical role in maintaining the stacking arrangement of the 2D layers. However, it was challenging to collect SCXRD data from samples exchanged with DMSO and EtOH due to crystal cleavage.

Therefore, to confirm the structure of the 2D layer itself using samples with different solvents, we conducted high-resolution transmission electron microscopy (HR-TEM) measurement. The sample preparation procedure was as follows: the samples were either washed in the mother liquid or subjected to solvent-exchanged with DMSO, DMF and EtOH, each for three cycles of 1 mL. Subsequently, the washed crystals were crushed through ultrasonication for 30 minutes. After ultrasonication, a droplet of the resulting suspension was transferred onto a carbon-film copper grid and allowed to dry for 12 hours in preparation for HR-TEM measurements. In the Fourier-filtered images of the as-synthesized, DMF, and EtOH samples, we observed a corrugated 2D layer in a folded form, except for the DMSO sample, which exhibited an amorphous phase. We measured d_1 and d_2 in the Fourier-filtered images of each sample and calculated the folding angle (θ_1) using a mathematical model. From this experiment, we identified two possibilities. Depending on the solvent, a more folded 2D layer can be obtained, as demonstrated in the EtOH sample. Furthermore, the degree of folding differs between single crystal and powder crystals, resembling a crystal size effect, see *Chem. Mater.* **32**, 4641–4650 (2020). However, the crucial point is that the 2D layer still retains the origami tessellation pattern.

We have included the effects of solvents on PPF-301 through HR-TEM analysis as follows on page 11, lines 11–21 of the “Conclusion” section in the manuscript, and a figure as shown in Supplementary Figure 21 (page 35 in the supplementary information).

“Generally, in 2D MOFs, solvents play a crucial role in maintaining the packing between 2D layers. Different types of solvents can affect the packing arrangement of these layers.^{33,63} In the case of the PPF-301 structure, the degree of folding in the 2D layer varies depending on the solvents used, in addition to the stacking pattern (Supplementary Fig. 21). Interestingly, the origami tessellation of the 2D layer, observed in Fourier-filtered images of HR-TEM, remains preserved regardless of the solvent. We observe that the inherent crumpled pattern of the 2D layer is maintained, suggesting that the folding mechanism remains the same regardless of the solvents, as the DCS pattern allows for one degree of freedom in deformation. Further research could be proposed by creating precisely controlled nanosheets to observe a more considerable movement of the 2D layer itself, to confirm the effect of the solvent and crystal size⁶⁴.”

b**Supplementary Figure 21. Origami tessellation analysis of samples exchanged with different solvents.**

a, HR-TEM images of the as-synthesized, DMF, and EtOH samples (scale bar: 20 nm), The corresponding fast Fourier-transform (FFT) patterns are inserted (scale bar: 1 nm^{-1}) in the upper figure. Fourier-filtered images with the PPF-301 structure superimposed (scale bar: 1 nm) in the lower figure. The as-synthesized PPF-301 crystals were prepared either by washing them in the mother liquid or by solvent-exchanging them with DMF and EtOH for $1 \text{ mL} \times 3$ times, respectively. The prepared crystals were subsequently crushed through ultrasonication for 30 min. After ultrasonication, a droplet of the resulting suspension was transferred onto a carbon-film copper grid and allowed to dry for 12 hrs in preparation for HR-TEM measurement. **b**, A comparison between experimental data obtained from HR-TEM measurements and a mathematical model. The relationships between θ_1 and θ_2 , as well as the relationships among lengths (d_1 and d_2) and θ_1 along the dotted line obtained from the mathematical model, were examined. The average values of d_1 and d_2 , measured from Fourier-filtered images of each sample, are marked on all of the graphs. θ_1 and θ_2 were calculated using Equation S1 and S2.

We have included as follows on page 14, lines 13–16 of the “Materials and Methods” section in the manuscript.

“**High-resolution transmission electron microscopy (HR-TEM) analysis.** HR-TEM analysis was performed with a JEM-2100 microscope (JEOL Company) equipped with a LaB₆ electron gun operated at an acceleration voltage of 200 kV. Generating Fast Fourier-Transform (FFT) patterns and image filtration were conducted by using Gatan DigitalMicrograph software.”

Reference added:

63. Ghosh, S. K. et al. A bistable porous coordination polymer with a bond-switching mechanism showing reversible structural and functional transformations. *Angew. Chem. Int. Ed.* **47**, 8843–8847 (2008).

64. Krause, S. et al. Impact of defects and crystal size on negative gas adsorption in DUT-49 analyzed by In Situ¹²⁹Xe NMR spectroscopy. *Chem. Mater.* **32**, 4641–4650 (2020).

(5) *What happens if the PPF-301 is desolvated (is that possible without structural disintegration?).*

A-5. Desolvation of PPF-301 results in structural disintegration since the solvent plays a role in maintaining the packing of the 2D layers. Following activation, PPF-301 experiences a loss of crystallinity and exhibits non-porosity in the N₂ isotherm graph as shown in Supplementary Figure 9 in the supplementary information.

Additionally, we compared crystal structures with or without solvent through the DFTB method. After conducting geometry and cell optimization without solvents, it was not possible to obtain a

structure that maintains the space group, and the cell contracts as the interlayer space without the solvent.

(6) If the mechanism were dependent on the solvated structure, and would disappear in the desolvated material how can the authors claim a very broad and fundamental impact of their discovery? Would this dependence limit the potential applicability of PPF-301?

In general, I am in favor of this manuscript, congratulate the authors to their discovery, and would recommend acceptance if the authors convincingly deal with the issues raised.

A-6. The corrugated 2D layer with a DCS pattern remained intact even after the packing of the 2D layers was altered through solvent exchange. TEM images obtained from solvent-exchanged samples clearly provided evidence of the inherent corrugated 2D layer in PPF-301. As highlighted in A-4, although the packing system of the 2D layers can be changed by solvent exchange, the inherent atomic arrangement of the 2D layers, which is compatible with the DSC pattern, remains preserved. Therefore, the origami mechanism based on folding movements is independent of the solvent. This outcome aligns with our assertion that the 2D layer of PPF-301 exhibits unparalleled properties as a metamaterial. Hence, the dependence on solvation does not limit the potential applicability of PPF-301, and its discovery holds broad and fundamental implications.

Reviewer #2 (Remarks to the Author):

In the work titled “Foldable Metal-Organic Framework as Origamic Mechanical Metamaterial”, Jin et al. report a kind of two-dimensional porphyrinic MOF which could display folding motions based on origami tessellation. By performing DFT calculations, the authors reveal the folding mechanism of a 2D porphyrinic sheet PPF-301 as changes in the dihedral and bond angles in the aryloxy group of the flexible linker. Even though the proposed idea of origami mechanical metamaterials is fascinating, however, most of the data were based on theoretical calculation, and no direct evidence (e.g., SEM or TEM photos) shows the flexible folding movement actually occurred on the 2D MOFs. After careful review and consideration, we believe this manuscript is more suitable for a specialized journal about computational chemistry instead of Nature Communications. Below are some detailed comments:

(1) *1. The authors should provide more experimental data to illustrate the proposed origamic flexible folding movement behavior of the 2D MOFs, such as scanning electron microscopy (SEM) or transmission electron microscopy (TEM) images directly showing the morphology changes.*

B-1. As suggested by reviewer 2, we conducted HR-TEM measurements of the PPF-301 structure using three different solvents (see also A-4 for details).

Additionally, SCXRD is a powerful measurement tool that provides direct atomistic evidence for structural transformations and reveals many interesting phenomena in nature. We acknowledge that crystallographic data serves as compelling evidence, enabling us to uncover precise movements of atoms and molecules. Although we measured a single crystal that includes stacked 2D layers, rather than a single isolated layer, we were able to demonstrate the structural transformation of the 2D layer itself through the analysis of folding movements as pivoting points, facilitated by the configuration change of the aryloxy group in the flexible porphyrinic linker.

(2) *2. The authors propose the 2D MOFs as mechanical metamaterials, however, no experimental results about the mechanical properties of the MOFs were found throughout the manuscript (including in the SI). X-ray diffraction and thermal analysis results are hard to reflect the intrinsic mechanical properties of MOFs, which makes it difficult to correlate the simulation results to the actual behavior of the materials.*

B-2. We attempted to conduct a pressure-induced single-crystal X-ray diffraction experiment on PPF-301 using a diamond anvil cell. We utilized a mixture of DMF/EtOH (1:1, v/v) as the pressure-transmitting medium. Initial measurements revealed a well-defined crystal structure with several clear and bright peaks. The data was analyzed using CrysAlis software, and the obtained cell parameters were in good agreement with the results of the SCXRD data.

	a (Å)	b (Å)	c (Å)	α (°)	β (°)	γ (°)
Initial measurement	11.09(5)	12.87(7)	14.05(5)	101.3(4)	100.5(3)	109.1(4)
SCXRD data	11.061(2)	12.842(3)	14.065(3)	100.77(3)	100.93(3)	109.11(3)

Our initial measurement confirmed the possibility of measuring the PPF-301 crystal under this experimental setup. However, it should be noted that these measurements represent an initial attempt, as our knowledge regarding the feasibility, solvent compatibility, suitable pressure ranges, and other factors was limited. Therefore, we believe that a separate study with higher accuracy and a more comprehensive and well-planned experiment should be conducted to fully understand the details of the structure transformation under pressure. This would warrant another paper to be published.

The pressure-induced SCXRD experiment was conducted in collaboration with Dr. Tobias Ritschel and Dr. Alexander Mistonov of Prof. Dr. Jochen Geck's group at the Institute of Solid State and Materials Physics. Additionally, Dr. Volodymyr Bon of Prof. Dr. Stefan Kaskel's group in the Department of Inorganic Chemistry at Technische Universität Dresden organized this collaboration.

We mentioned their support in the Acknowledgments.

(3) 3. *In the manuscript, according to the DFT calculation, how can origami tessellation improve the properties of MOF to make it a mechanical metamaterial?*

B-3. Through DFT calculations, it was found that the 2D layer of PPF-301 exhibits a remarkable mechanical property known as a negative Poisson's ratio (NPR), which is a characteristic feature of mechanical metamaterials. NPR materials become thicker in the vertical direction when subjected to horizontal strain. The NPR of the 2D layer of PPF-301 was investigated using DFT calculation. The atomic movements associated with achieving the minimal Poisson's ratio are consistent with changes in the dihedral angle and bond angle of the aryloxy group, which are the main factors contributing to the origami-like movement. The dihedral angle and bond angle influence the folding angle, which drives the origamic movement. Therefore, the incorporation of origami tessellation into the MOF structure imparts mechanical metamaterial properties to the material.

In principle, materials designed using origami tessellation are known to possess exceptional properties as mechanical metamaterials, as discussed in the "Introduction" section.

See: "For example, the DCS and square twist patterns exhibit the same repeating patterns, but their folding movement differs. Both tessellations are highly deployable^{29,30} and can serve as a blueprint for constructing mechanical metamaterials with negative Poisson's ratio which is well-known for an exotic mechanical property.^{31,32}"

(4) *What is the relationship between the folding angles and mechanical properties of MOF?*

B-4. As explained in B-3, the origami-like movement is triggered by the folding-unfolding process. During this process, the folding angle between each tile and its adjacent tile undergoes continuous changes. As the folding angle increases, resulting in the unfolding of the crumpled 2D layer, we observe a corresponding increase in the 2D area which we have arbitrarily defined this area with a dotted line, as shown in the figure below.

See Supplementary Figure 15b.

An interesting aspect is that when one side of the 2D layer is stretched, it expands in the direction perpendicular to that side in the same manner. Conversely, when one side is contracted, it uniformly

shrinks. This property is unique to materials with origami tessellation and is not easily observed in normal 2D layers. These properties can be associated with the mechanical property known as NPR property. Thus, the origami-like movement induced by the changes in the folding angle leads to the NPR property, which is one of the mechanical properties.

(5) 4. Is DCS origami tessellation applicable to most two-dimensional MOFs, and what are the applicable conditions?

B-5. The concept of DCS origami tessellation can be effectively employed in the precise design of PPF-301 with inherent flexibility, distinguishing it from the majority of two-dimensional MOFs. To meet the applicable conditions, the flexibility of the 2D MOFs is crucial. This can be achieved by incorporating flexible linkers within the structure, rather than relying solely on the flexible motion between 2D layers, such as interlayer expansion or contraction. Unfortunately, a significant portion of the reported 2D MOFs lacks flexibility within the 2D layer itself. To address this limitation, it is essential to focus on designing the central flexible region of 2D MOFs using diverse origami tessellations, including DCS patterns.

Reviewer #3 (Remarks to the Author):

This paper describes a mechanistic insight into the flexibility of a two-dimensional metal-organic framework (MOF). The authors found one reported two-dimensional MOF, here named PPF-301 (Goldberg et al. CrystEngComm 2010, 4095.), assembled from zinc ions and porphyrin ligand (TCMOPP), shows Origami-like tessellation in response to temperature. This “paper folding” behavior of two-dimensional MOF was further structurally correlated to the specific angle of chemical bonds. Indeed, this paper is similar to the work done by the same authors (Metal-organic framework based on hinged cube tessellation as transformable mechanical metamaterial, Sci. Adv. 2019, eaav4119). The previous Sci Adv paper discussed similar tessellation behavior based on three-dimensional MOF. Even though I see a certain novelty in this paper such as 2D MOF and paper-folding tessellation mechanism, what the authors demonstrated experimentally, such as temperature-dependent single crystal X-ray diffraction experiments, looks routine. I would happily accept this paper if the authors could have shown another stimulus to induce the structural change. At this moment, I am not so convinced that this paper showed the novelty criterion for publication in Nature Communications. The more detailed scientific concerns and discussions are as follows.

(1) *1. Both this and previous papers discussed the mechanical metamaterial behavior of MOF only by structural change in response to temperature. Of course, the positive or negative thermal expansion of the framework should be first investigated; however, I was expecting that another stimulus could change the structure of this system because the authors discovered such metamaterial-like behavior already in the previous paper. For instance, pressure-dependent SCXRD measurement could have been carried out to see the real effect of mechanical stress on structural change.*

C-1. As a response to the reviewer’s comment, we attempted to measure the pressure-induced SCXRD of PPF-301. Please refer to section B-2 for more details.

Furthermore, both PPF-301, based on origami tessellation as presented in this work, and UPF-1, employing cube tessellation as reported in previous work (*Sci. Adv.* eaav4119 (2019)), exhibit common metamaterial-like behavior. Although the overarching concept of “Meta-MOFs” remains the same in these studies, the specific molecular transformations and underlying mechanisms are significantly different. One notable distinction from the previous paper lies in the fascinating aspect of structural analysis within the 2D framework based on origami tessellation. Origami tessellation, starting from a two-dimensional material, enables the creation of a three-dimensional material. Essentially, both papers highlight the significance of introducing novel molecular arrangements to develop Meta-MOFs within the field of MOFs. The study and exploration of internal arrangements, which have been extensively investigated for various metamaterials, are continuously being applied to the design and analysis of MOF structures to create Meta-MOFs. This is because the unprecedented properties of metamaterials originate from the arrangements themselves, regardless of the specific components and size.

(2) *2. Another way to improve this paper would be generalizability. By understanding this mechanism, the authors might be able to suggest any other molecular system that can show this origami tessellation. If the authors are able to design such a new ligand or metal node and demonstrate it, this paper will become more robust and more appealing.*

C-2. We propose the development of new origamic 2D MOFs by embedding flexibility into the structure, which is crucial for introducing origami tessellation. One approach is to design different functionalized linkers that can incorporate flexibility within the 2D framework. For example, in the case of the PPF-301 structure, the porphyrinic linker with an aryloxy group plays a vital role in

determining the folding angles. By substituting the aryloxy group with other functional groups such as $-CH_2-$, $-S-$, and $-NH-$, we anticipate different folding motions. Each functional group will have its potential energy surface corresponding to changes in configuration, including folding and unfolding. We have performed calculations on the potential energy surface of an isolated N-phenylglycine molecule with the $-NH-$ functional group. Based on this calculation, we anticipate that a new 2D MOF could be created in an unfolded state. This series opens up the possibility of creating a new stream of 2D MOFs, including metal paddlewheel clusters with variations such as Cu, Ni, and Co.

We add perspective about new 2D organic MOFs in the “Conclusion” section (page 12, lines 2–5 in manuscript) and Supplementary Figure 22 on page 36 in the supplementary information.

“Otherwise, to alter the flexibility of the 2D layer itself for new organic MOFs, we can explore the incorporation of diverse functional groups such as $-CH_2-$, $-S-$, and $-NH-$ instead of the aryloxy group. The preferred dihedral and bond angles associated with each functional group act as pivot points, leading to varying degrees of folding movement (Supplementary Fig. 22).”

Supplementary Figure 22. Potential origamic MOFs by changing the functional groups of a porphyrinic linker. **a**, The porphyrinic linker can be substituted by various functional groups; $-O-$, $-CH_2-$, $-S-$, and $-NH-$. **b**, Potential energy surface of N-phenylglycine ($-NH-$) molecule by varying φ and α . This molecule can be stabilized in its more folded conformation than the aryloxy group.

(3) 3. After showing the correlation between the molecular structure and the simplified schematic model in Figure 2, there is no molecular model in the figure. This makes it more difficult to understand how much the structure changed in response to temperature. It would be better to show and overlay two extreme cases of origami tessellation for clarity.

C-3. We have included a new figure in Supplementary Figure 13, which overlays two crystallographic structures at 100 K (blue color fragment) and 380 K (red color fragment) as shown on page 22 in the supplementary information.

Supplementary Figure 13. Overlaid crystallographic structures at 100 K (blue color fragment) and 380 K (red color fragment).

We have added the following statement on page 7, lines 5–8 of the manuscript.

“Notably, as the area S expands, the thickness of the layer (d_3) decreases by 2.6 %, which is similar to the principles of origami mechanics, where overlaid molecular structures at 100 K and 380 K aid in understanding the molecular movement (Fig. 2b, c and Supplementary Fig. 13).”

Reviewer #4 (Remarks to the Author):

This manuscript proposes a new idea to use foldable MOF as origami architected metamaterial, which provides a new approach for designing and building mechanical materials at molecular level, with adjustable properties. The concept is novel and interesting, but the execution seems not comprehensive and fully convincing, despite that the authors conducted a thorough experimental investigation of the MOF mechanical properties complemented with computational modelling to demonstrate the rational design of various MOF-based architectures. The experimental part (including manufacturing and characterization) and actual demonstration should be substantially enhanced.

Some specific comments to be addressed as well:

(1) *1. Based on the literature, there are some general work principles and applications of metal-organic frameworks (MOFs) at the molecular level. However, in the intro section, the author describes the origami structure in detail (well done), which is not however very friendly to those who are unfamiliar with MOFs. What are the uniqueness and advantages of the proposed approach compared with previous work? The author did mention parts of that in lines 56-62. Nevertheless, it is not clear to me. The author may need to state them clearly.*

D-1. We revised the paragraph as referred to by the reviewer from page 3, line 19 to page 4, line 5 of the manuscript.

“To create origami-inspired materials at the molecular level, metal-organic frameworks (MOFs) could serve as an ideal platform for mimicking origami patterns, thanks to the unique features that the building blocks, metal nodes and organic linkers, used for MOF construction are virtually limitless and exquisitely tunable.^{33,34} Through rational design based on deformable net topology, many MOFs have exhibited structural flexibility, derived from the inherent flexibility of their structural building blocks over the past two decades.³⁵⁻³⁷ The rich structural choices serve to realize the deployable two-dimensional (2D) framework itself, showing an unprecedented property like a negative thermal expansion.³⁸ While the predictable deployable movement of these flexible MOFs demonstrates remarkable mechanical properties with the metamaterials,³⁹⁻⁴² a new geometrical analysis involving origami tessellation to uncover hidden dynamic motions in MOFs beyond typical topological analysis is still in its infancy⁴³.”

References added:

39. Coudert, F.-X. & Evans, J. D. Nanoscale metamaterials: Meta-MOFs and framework materials with anomalous behavior. *Coord. Chem. Rev.* **388**, 48–62 (2019).

40. Xing, Y. et al. Exploration of hierarchical metal-organic framework as ultralight, high-strength mechanical metamaterials. *J. Am. Chem. Soc.* **144**, 4393–4402 (2022).

41. Evans, J. D., Bon, V., Senkovska, I., Lee, H.-C. & Kaskel, S. Four-dimensional metal-organic frameworks. *Nat. Commun.* **11**, 2690 (2020).

(2) *2. The author aim to implement a design approach of origami mechanical metamaterial, but the optimization analysis of the structure towards some functionality, e.g., reconfigurability of the structure, or specific stiffness beyond ensuring the foldability of the origami structure is missing.*

D-2. To address the reconfigurability aspect of PPF-301, we conducted temperature-dependent SCXRD experiments. The crystal structure of PPF-301 remained intact without any loss of crystallinity as the temperature decreased, demonstrating crystallographic reversibility (100 K → 380 K → 220 K).

(3) At the moment, the authors discuss the negative thermal expansion (NTE) of 2D layer, but the discussion of functionality of the original structure is lost (i.e., foldability). What is specifically the objective function in their approach, and what are the constraints?

D-3. In its original structure, PPF-301 consists of stacked 2D layers. Since directly measuring the structural transformation of a single 2D layer using SCXRD under external stimuli was challenging, we utilized the original structure to investigate the mechanical movement based on origami tessellation. Additionally, we focused on analyzing the behavior of a single 2D layer to enhance the reader's understanding of the origami movement. During the flattening of the corrugated 2D layer, the area (S) expands by 2.0 % while the thickness of the 2D layer decreases by 2.6 %. While the negative thermal expansion (NTE) of the thickness influences the transition of the cell parameter, the overall cell parameter increases due to a larger expansion of the interlayer spacing (3.1 %) between the 2D layers.

We have revised the paragraph from page 6, line 18 to page 7, line 12 of the manuscript.

“Firstly, we note an interesting change in the cell parameters of PPF-301 in a temperature range of 100–380 K. As the temperature decreases from 380 to 220 K, the cell parameters exhibit complete reversibility without hysteresis (Supplementary Fig. 11). The cell volume progressively increases by 5.2 % upon heating, accompanied by changes in the a and b parameters, as well as the γ value (Supplementary Fig. 12). To analyze such structural changes in detail, we focus on the 2D area (S) and interlayer spacing of PPF-301. The area S and the interlayer spacing increase by 2.0 % and 3.1 %, respectively (Fig. 2a and Supplementary Table 2). The expansion of S and interlayer spacing contributes to the increase in cell volume. While the change in interlayer spacing is commonly observed in 2D MOFs, the change in the 2D layer itself is rather exceptional.³⁸ Notably, as the area S expands, the thickness of the layer (d_3) decreases by 2.6 %, which is similar to the principles of origami mechanics, where overlaid molecular structures at 100 K and 380 K aid in understanding the molecular movement (Fig. 2b, c and Supplementary Fig. 13). The 2D layer exhibits negative thermal expansion (NTE) as the thickness shrinks. The NTE of the thickness influences the transition of the cell volume, but the overall cell volume increases due to a larger expansion of the interlayer spacing between the 2D layers. The NTE property of the 2D layer in MOFs is significantly rare because most flexible 2D MOFs experience transformation in the interlayer.⁴⁷”

(4) 3. *The authors did not discuss the scalability of these MOF-based metamaterials, the feasibility of metamaterials under subsidy parameters is verified in the Supplementary materials, but there is no further study of the versatility of other origami structures and materials, and there is no in-depth discussion of the manufacturability, material cost, and time to produce these metamaterials, which are important (otherwise would be just some MOF folding simulation)*

D-4. The concept of “Meta-MOFs” was first introduced in 2019 (*Coord. Chem. Rev.* **388**, 48–62 (2019), *Sci. Adv.* eaav4119 (2019)). Since then, fascinating research on “Meta-MOFs” has been conducted (*J. Am. Chem. Soc.* **144**, 4393–4402 (2022), *Trends Chem.* **3**, 254–265 (2021), *Nat. Commun.* **11**, 2690 (2020)). However, the field of MOF-based metamaterials is still in its infancy. In particular, the meta-MOF based on origami tessellation presented in this work is the first example of its kind. Therefore, it is challenging to discuss commercial aspects such as material cost and production time at this stage. Nonetheless, we provide a comprehensive discussion on the designability of new origamic MOFs and the effects of external stimuli, such as solvents, on the origami structure. This extends the scope of origamic MOFs as metamaterials in conclusion (Please see A-4 and C-2 for details).

(5) 4. *While the article briefly mentions several potential applications for MOF-based metamaterials, such as flexible and mobile electronics, self-adaptive sensor systems, and soft robotics. However, there was no discussion regarding the practicality, feasibility, and challenges of utilizing these metamaterials in real-world applications, especially on what specific applications this proposed origamic mechanical metamaterial would be best for, and why the specific capabilities of the MOF are critical to enable such applications.*

D-5. We acknowledge that the applications mentioned in the “Introduction” section pertain to practical applications of macroscopic materials with origami tessellation, rather than MOF-based metamaterials.

As stated in D-4, this work represents a milestone as the first example of Meta-MOF induced by molecular movement based on origami tessellation. While real-world applications are yet to be explored, it is important to suggest their potential applications. Therefore, we propose one potential

application related to 2D paddlewheel MOFs as 2D spin qubit frameworks for the development of advanced molecular quantum computing as discussed in the “Conclusion” section.

See: “Furthermore, the folding movement based on origami tessellation shown here allows controlling the distance between metal nodes upon external stimuli, which could potentially provide 2D spin qubit frameworks^{65,66} to develop advanced molecular quantum computing— one of the future applications remaining for origamic MOFs.”

Some minor issues:

(6) 1. The temperature unit should be uniform. Line 79 is Celsius; the rest is Kelvin.

D-6. The temperature unit for TGA data has been changed from Celsius to Kelvin.

“PPF-301 is thermally stable up to ~700 K and non-porous to N₂ at 77 K (Supplementary Fig. 8 and 9).” (Page 5, line 24 – page 6, line 1 in the manuscript)

“**Thermogravimetric analysis (TGA).** TGA was conducted using a TA instrument SDT Q600, with heating performed from 303 K to 1073 K under an N₂ atmosphere at a scan rate of 10 K min⁻¹.” (Page 14, lines 3–5 in the manuscript)

Supplementary Figure 8. Thermogravimetric analysis data for PPF-301. The initial weight loss (~27 %, 300–420 K) corresponds to solvent removal, followed by decomposition of the structure at around 700 K (~40 % weight loss).” (Page 14 in the supplementary information)

(7) 2. As an original research, the Fig. 1 appears not very necessary...

D-7. Fig.1 in the manuscript has been moved to Supplementary Figure 1 in the supplementary information (page 3 in the supplementary information). We have retained this figure to provide a summary of the history of origami tessellation and related materials.

REVIEWER COMMENTS

Reviewer #1 (Remarks to the Author):

The authors did a very good job in performing significant new experiments and adding new data and explanations to the questions and issues raised during the review. This reviewer is satisfied with the changes made and the discussion and arguments provided.

Still, however, it may remain a controversy about the over-all novelty and the generalization of the concepts in order to obtain functional MOF-Metamaterials (including responsiveness to a broader spectrum of stimuli). On balance, this reviewer is in favor of accepting the manuscript as it is now.

Reviewer #2 (Remarks to the Author):

In the revised manuscript, the authors extensively revised the manuscript with many new data incorporated to address the reviewers' concerns. The overall quality of the revised manuscript has been improved a lot even though they haven't provided the SEM or TEM images showing this folding-unfolding dynamic origami behavior as I requested. The newly incorporated HRTEM data in Supplementary Figure 21 shows PPF-301's response to different solvents instead of its response to temperature changes, which could be done by utilizing the advanced in situ TEM characterization techniques. However, if the authors insist that the SCXRD is powerful enough to reveal the precise movements of atoms and molecules, the original XRD patterns from which the results listed in Figure 2c), Figure 3b-c), and Supplementary Figure 11-12 were calculated out are required. Below are the itemized concerns regarding the revised manuscript:

1. Please provide the original SCXRD patterns for the results listed in Figure 2c), Figure 3b-c), Supplementary Figure 11-12, and others as long as they are experimental data depicting the dynamic changes of lattice parameters, cell volumes, etc. Then please briefly illustrate how the lattice parameters were calculated based on those XRD patterns.
2. In the cubic crystalline system, the plane family of {111} are the closest packed planes. Are {111} also the closest packed planes for PPF-301 which belongs to the Triclinic crystal system as the authors presented in Supplementary Figure 5 and Supplementary Table 1?
3. On lines 141-143 of page 7, the authors wrote "The layer in PPF-301 has four different types of tiles, labeled as A, B, C, and D with colors such as yellow, blue, gray, and dark gray, respectively." Even though they have labeled the ABCD clearly in Supplementary Figure 16, to facilitate the precise understanding of the readers, here I suggest also marking the ABCD tiles onto the models presented in Figure 2a-b) in accordance with Supplementary Figure 16.
4. On lines 176-178 of page 9, what's the mechanism driving the changes of dihedral angle and bond

angle of the aryloxy group upon thermal stimuli? Could it be generalized to other MOFs containing the aryloxy groups?

5. The Fourier-filtered images (Supplementary Figure 21) showed that the structural differences of 2D MOF in different solvents may depend on different orientations, and may not reflect “the folding-unfolding process in the origami-like movement”.

6. The author’s attempt to use a diamond anvil cell to conduct a pressure-induced single-crystal X-ray diffraction experiment on PPF-301 is worthy of recognition, the results will be more convincing if pressure values are provided. In addition, PPF-301 can remain stable in DMF or EtOH solvent, is it effective to use a mixture of DMF/EtOH as the pressure-transmitting medium?

7. According to the authors’ reply, “the atomic movements associated with achieving the minimal Poisson’s ratio are consistent with changes in the dihedral angle and bond angle of the aryloxy group...” and “solvents can affect the degree of folding...”, is it possible to consider that PPF-301 has a greater Poisson’s ratio span in other solvents?

Reviewer #3 (Remarks to the Author):

The revised manuscript certainly improved the quality of the manuscript. The authors added more experimental data to support their claim. Particularly, the HR-TEM experiments are quite convincing. Though the authors were not able to demonstrate the structural change in response to mechanical stress due to its technical difficulty, I would rather support the acceptance of this manuscript in Nature Communications.

My remaining concern is the title “Foldable Metal-Organic Framework as Origamic Mechanical Metamaterials”. This title sounds exaggerated, compared to the exact structural transformation the authors presented in the manuscript. This is because of the following two reasons.

(1) The term “foldable” gives us the impression that the 2D MOF sheet is really folded; one part of the sheet is touching another part of the sheet. Or at least the bending for the valley fold should be acute angle. However, as the authors mentioned in the answer to my previous comment, the structural change is very small (less than a few percent). It is hard to call such structural motion “foldable”.

(2) The term “Origamic Mechanical Metamaterials” also gives the impression that the 2D MOF sheet is folded by mechanical stress. Indeed, the authors only demonstrated the structural change in response to temperature. As the authors answered in my previous comment, the authors failed to demonstrate the high-pressure crystallography for this MOF.

From these points, I feel that the authors try to oversell the concept. Indeed, the Figure 5C is also misleading. The PPF-301 does not show such a large structural deformation. The authors should change the title to fit the more realistic motion of this 2D MOF.

Reviewer #4 (Remarks to the Author):

The authors have addressed most of my technical comments and made substantial improvements in their manuscript, to better highlight the novelty of the concept as well as the general applicability of their results.

It is however suggested that the author should include some of the actual sample/experiment results into the main manuscript figures, rather than just shown in Supplementary Information.

REVIEWER COMMENTS

Reviewer #1 (Remarks to the Author):

The authors did a very good job in performing significant new experiments and adding new data and explanations to the questions and issues raised during the review. This reviewer is satisfied with the changes made and the discussion and arguments provided.

Still, however, it may remain a controversy about the over-all novelty and the generalization of the concepts in order to obtain functional MOF-Metamaterials (including responsiveness to a broader spectrum of stimuli). On balance, this reviewer is in favor of accepting the manuscript as it is now.

1.1. Thanks for your constructive input on this manuscript.

Reviewer #2 (Remarks to the Author):

In the revised manuscript, the authors extensively revised the manuscript with many new data incorporated to address the reviewers' concerns. The overall quality of the revised manuscript has been improved a lot even though they haven't provided the SEM or TEM images showing this folding-unfolding dynamic origami behavior as I requested. The newly incorporated HRTEM data in Supplementary Figure 21 shows PPF-301's response to different solvents instead of its response to temperature changes, which could be done by utilizing the advanced in situ TEM characterization techniques. However, if the authors insist that the SCXRD is powerful enough to reveal the precise movements of atoms and molecules, the original XRD patterns from which the results listed in Figure 2c), Figure 3b-c), and Supplementary Figure 11-12 were calculated out are required. Below are the itemized concerns regarding the revised manuscript:

1. Please provide the original SCXRD patterns for the results listed in Figure 2c), Figure 3b-c), Supplementary Figure 11-12, and others as long as they are experimental data depicting the dynamic changes of lattice parameters, cell volumes, etc. Then please briefly illustrate how the lattice parameters were calculated based on those XRD patterns.

2.1. All of the SCXRD crystallographic data were already provided with a *checkcif* file (checkcif_100 K to 380 K, totaling 15 files) and *cif* file (PPF-301_100 K to 380 K, totaling 15 files) at the submission stage.

The crystallographic values (*a*, *b*, *c*, α , β , γ , and *V*) related to Fig. 2c, 3b–c, and Supplementary Fig. 11–12 can be found in Supplementary Table 1.

Additionally, the calculation method has already been described in the Materials and Methods (page 13 in the Manuscript).

2. In the cubic crystalline system, the plane family of $\{111\}$ are the closest packed planes. Are $\{111\}$ also the closest packed planes for PPF-301 which belongs to the Triclinic crystal system as the authors presented in Supplementary Figure 5 and Supplementary Table 1?

2.2. The plane for triclinic PPF-301 is $\{\bar{1}11\}$.

3. On lines 141-143 of page 7, the authors wrote "The layer in PPF-301 has four different types of tiles, labeled as A, B, C, and D with colors such as yellow, blue, gray, and dark gray, respectively." Even though they have labeled the ABCD clearly in Supplementary Figure 16, to facilitate the precise understanding of the readers, here I suggest also marking the ABCD tiles onto the models presented in

Figure 2a-b) in accordance with Supplementary Figure 16.

2.3. According to the reviewer's suggestion, we add ABCD tiles in Fig. 2a on page 24 of the Manuscript.

Fig. 2. Thermal response of PPF-301. **a**, Four types of different-sized tiles, filled in the 2D sheet. A; Zn SBU (yellow), B; porphyrinic ligand (blue), C and D; hollow tiles (gray and dark gray, respectively). **b**, 2D area (S) of the PPF-301. Area S is defined by connecting each centroid of Zn SBUs ($S = d_1 \times d_2 \times \sin \sigma$). **c**, Thickness (d_3) of the 2D corrugated layer. **d**, S and d_3 as a function of temperature from 100 to 380 K.

4. On lines 176-178 of page 9, what's the mechanism driving the changes of dihedral angle and bond angle of the aryloxy group upon thermal stimuli? Could it be generalized to other MOFs containing the aryloxy groups?

2.4. As shown in Fig. 4c, isolated aryloxy group found in the CSD represents variations in dihedral angle of about $60\text{--}90^\circ$ and bond angle of about $115\text{--}129^\circ$. Such flexible aryloxy group can exhibit significant variation and highly sensitive to external stimuli like temperature. We expect similar structural flexibility in other MOFs with the aryloxy group when appropriate external stimuli are given.

5. The Fourier-filtered images (Supplementary Figure 21) showed that the structural differences of 2D MOF in different solvents may depend on different orientations, and may not reflect "the folding-unfolding process in the origami-like movement".

2.5. The corresponding issue was a prerequisite for distinguishing the folding state by HR-TEM. The TEM data represent the folding state for the following reasons. First, to avoid any potential for misleading results, we captured several TEM images for each batch of samples under our solvent conditions, repeating the process at least three times to acquire the standard deviations. The average values of 'd' values exhibited clear difference from one another. Second, 2D MOFs, including PPF-301, exhibit highly anisotropic structural features, particularly in terms of the strength of interlayer and interlayer interactions. Therefore, in many cases, the nanosheets indicated the zone axis perpendicular to the sheet plane as demonstrated by preferential orientation in PXRD patterns.

6. The author's attempt to use a diamond anvil cell to conduct a pressure-induced single-crystal X-ray diffraction experiment on PPF-301 is worthy of recognition, the results will be more convincing if pressure values are provided. In addition, PPF-301 can remain stable in DMF or EtOH solvent, is it effective to use a mixture of DMF/EtOH as the pressure-transmitting medium?

2.6. To conduct a pressure experiment under the same sample conditions as the temperature experiment, a mother liquid (the mixture of DMF/EtOH) was used.

7. According to the authors' reply, "the atomic movements associated with achieving the minimal Poisson's ratio are consistent with changes in the dihedral angle and bond angle of the aryloxy group..." and "solvents can affect the degree of folding...", is it possible to consider that PPF-301 has a greater Poisson's ratio span in other solvents?

2.7. It is possible to obtain different negative Poisson's ratios because the initial degree of folding strongly depend upon the solvent used.

Reviewer #3 (Remarks to the Author):

The revised manuscript certainly improved the quality of the manuscript. The authors added more experimental data to support their claim. Particularly, the HR-TEM experiments are quite convincing. Though the authors were not able to demonstrate the structural change in response to mechanical stress due to its technical difficulty, I would rather support the acceptance of this manuscript in Nature Communications.

My remaining concern is the title "Foldable Metal-Organic Framework as Origamic Mechanical Metamaterials". This title sounds exaggerated, compared to the exact structural transformation the authors presented in the manuscript. This is because of the following two reasons.

(1) The term "foldable" gives us the impression that the 2D MOF sheet is really folded; one part of the sheet is touching another part of the sheet. Or at least the bending for the valley fold should be acute angle. However, as the authors mentioned in the answer to my previous comment, the structural change is very small (less than a few percent). It is hard to call such structural motion "foldable".

(2) The term "Origamic Mechanical Metamaterials" also gives the impression that the 2D MOF sheet is folded by mechanical stress. Indeed, the authors only demonstrated the structural change in response to temperature. As the authors answered in my previous comment, the authors failed to demonstrate the high-pressure crystallography for this MOF.

From these points, I feel that the authors try to oversell the concept. Indeed, the Figure 5C is also misleading. The PPF-301 does not show such a large structural deformation. The authors should change the title to fit the more realistic motion of this 2D MOF.

3.1. According to the Reviewer 3's suggestion, we have modified the title to "Origamic Metal-Organic Framework toward Mechanical Metamaterial". This MOF is an NPR material, as supported by DFT calculation. It is well-established that NPR material is classified as a member of mechanical metamaterials (see ref. 32).

Regarding Fig. 5C, we do not intend to mislead the reader. Fig. 5C helps the readers to visualize the transformation of DCS origami found in PPF-301.

Reviewer #4 (Remarks to the Author):

The authors have addressed most of my technical comments and made substantial improvements in their manuscript, to better highlight the novelty of the concept as well as the general applicability of their results.

It is however suggested that the author should include some of the actual sample/experiment results into the main manuscript figures, rather than just shown in Supplementary Information.

4.1. The authors would like to thank Reviewer 4 for constructive suggestion, moving some of the figures to manuscript. We have rearranged the HR-TEM data regarding to the solvent effect to Fig. 6 of the Manuscript.

REVIEWERS' COMMENTS

Reviewer #2 (Remarks to the Author):

As long as the crystallographic data presented in this work, which effectively demonstrate the intriguing dynamic origami process of the 2D MOFs, were obtained through experimental measurements, this work can be accepted for publication.

REVIEWER COMMENTS

Reviewer #2 (Remarks to the Author):

As long as the crystallographic data presented in this work, which effectively demonstrate the intriguing dynamic origami process of the 2D MOFs, were obtained through experimental measurements, this work can be accepted for publication.

2.1. Thank you for your constructive comments and for accepting this manuscript.